# Lack of peroxisomal catalase affects heat shock response in *Caenorhabditis elegans*

Marina Musa[1], Pedro A Dionisio[2], Ricardo Casqueiro[2], Ira Milosevic[2,3], Nuno Raimundo[2,4], Anita Krisko[5]

**Exact mechanisms of heat shock–induced lifespan extension, although documented across species, are still not well understood. Here, we show that fully functional peroxisomes, specifically peroxisomal catalase, are needed for the activation of canonical heat shock response and heat-induced hormesis in *Caenorhabditis elegans*. Although during heat shock, the HSP-70 chaperone is strongly up-regulated in the WT and in the absence of peroxisomal catalase (*ctl-2(ua90)II*), the small heat shock proteins display modestly increased expression in the mutant. Nuclear foci formation of HSF-1 is reduced in the *ctl-2(ua90)II* mutant. In addition, heat-induced lifespan extension, observed in the WT, is absent in the *ctl-2(ua90)II* strain. Activation of the antioxidant response and pentose phosphate pathway are the most prominent changes observed during heat shock in the WT worm but not in the *ctl-2(ua90)II* mutant. Involvement of peroxisomes in the cell-wide cellular response to transient heat shock reported here gives new insight into the role of organelle communication in the organism's stress response.**

## Introduction

Heat shock response (HSR) is the cellular response to stress, characterized by robust up-regulation of heat shock proteins (HSPs) or chaperones under control of transcription factor HSF-1 in *Caenorhabditis elegans*. It is necessary for maintenance of proteostasis, disruption of which is associated with several human diseases, including Parkinson's, Alzheimer's, and even some forms of cancer (Morimoto, 2008; Mendillo et al, 2012). The adaptive stress response in recent years has been studied in the context of aging and disease, and many hormetic treatments have been demonstrated to be beneficial for survival. These beneficial effects have generally been attributed to the activation of stress response pathways, which depend on the type of stressor and downstream signaling targets (Cypser & Johnson, 2002; Calabrese et al, 2007; Rattan & Demirovic, 2009).

Although the connection between stress and hormesis is well described, the exact mechanisms are still under debate. Canonically, hormetic effects were attributed to the increased concentration of chaperones inside the cell, but it is now accepted that a more complex scenario takes place. Several pathways aside from HSR have been demonstrated to be needed for hormetic effect to occur, including induction of autophagy, inhibition of TORC1, increase in respiration, and accumulation of mitochondrial ROS (Olsen et al, 2006; Bonawitz et al, 2007; Pan & Shadel, 2009; Van Raamsdonk & Hekimi, 2009; Yang & Hekimi, 2010; Kumsta et al, 2017; Perić et al, 2017; Musa et al, 2018).

Peroxisomes, oxidative organelles that play a role in oxidation of very long fatty acids, have a number of vital roles that may affect survival during stressful conditions, including but not limited to their connection with mitochondria, are already implicated in many aspects of stress biology. Reactive oxygen species produced by peroxisomes are a major contributing factor to oxidative stress, which is implicated in aging, aging-driven diseases, and cell death. Peroxisomes also display their own oxidative stress response to ROS produced both in peroxisomes and elsewhere in the cell. Yet, the involvement and potentially essential role of peroxisomes in the HSR have not been explored at cellular and organismal level. Here, we present data suggesting that the peroxisomal catalase plays a crucial role for the HS-induced lifespan extension in *C. elegans* and possibly HSR activation. Animal peroxisomes have not been much discussed in the context of stress and hormesis despite being hubs of ROS and fatty acid metabolism. Our results show that the peroxisomal catalase may be required for proper heat shock–induced stress response and hormesis to occur, thus suggesting a crosstalk between the oxidative stress and heat shock in *C. elegans*.

## Results

### Peroxisomal mutants have altered HSR and impaired thermotolerance

We set to investigate the metabolic response of *C. elegans* to mild HS and a putative role of peroxisomes in this process. We focused

[1]Mediterranean Institute for Life Sciences, Split, Croatia   [2]Multidisciplinary Institute of Ageing, University of Coimbra, Coimbra, Portugal   [3]Nuffield Department of Medicine, Wellcome Centre for Human Genetics, NIHR Oxford Biomedical Research Centre, University of Oxford, Oxford, UK   [4]Department of Cellular and Molecular Physiology, Penn State University College of Medicine, Hershey, PA, USA   [5]Department of Experimental Neurodegeneration, University Medical Center Göttingen, Göttingen, Germany

Correspondence: krisko.anita@gmail.com

on two *C. elegans* strains: WT and peroxisomal mutant *ctl-2(ua90)II*. The *ctl-2(ua90)II* mutant, which carries a 1,060 bp deletion in 5' region of the peroxisomal catalase gene *ctl-2*, was chosen because it displays a shortened lifespan (Petriv & Rachubinski, 2004). Peroxisomal catalase *ctl-2* is the only peroxisomal catalase and is responsible for about 80% of all catalase activity in *C. elegans* (Petriv & Rachubinski, 2004). The 4-h HS at 30°C was administered in L4 stage, whereas control worms were kept at 20°C throughout the experiment (optimal growth temperature, OGT). We chose the late L4 stage as opposed to adults because worms are mostly developed at L4 and have no embryos, which avoids potentially confounding effects, especially in the assessment of transcript levels, because they may express HSPs because of their role in developmental processes as opposed to heat stress. Furthermore, the L4 stage is the most sensitive to HS in terms of HSR, although exhibiting higher survival rate than younger or older worms after HS (Zevian & Yanowitz, 2014). For WT worms, we measured increased median and maximum lifespan; median lifespan was increased from 11 to 14 d post HS (≈20% increase) and maximum lifespan from 17 to 20 d (15% increase) (Fig 1A). In contrast, although maximum lifespan of the *ctl-2(ua90)II* strain after HS was increased from 13 to 15 d (≈13% increase), the median lifespan was unchanged and measured as 10 d for both HS and OGT *ctl-2(ua90)II* worms (Fig 1B). Overall, mild HS extended the lifespan of WT but did not significantly affect *ctl-2(ua90)II* strain lifespan.

As robust up-regulation of chaperones/HSPs is the major hallmark of HSR, we measured the transcript levels of several HSPs. Surprisingly, we saw that although small HSPs (sHSPs) are robustly up-regulated in WT during mild HS, their up-regulation is not as robust in the *ctl-2(ua90)II* mutant (Fig 1C and D). Although in WT, HSP-16.1 and HSP-16.2 undergo 350-fold and 1,700-fold up-regulation at transcript level, in the *ctl-2(ua90)II* mutant, they increase only 40-fold and 70-fold, respectively. Interestingly, the expression of Hsp16.2 in young worms in response to stress has been positively correlated with lifespan (Rea et al, 2005). On the other hand, transcript levels of HSP-70 are almost twice as abundant in the *ctl-2(ua90)II* mutant compared with WT (Fig 1E), further suggesting that the transcriptional response to HS in the *ctl-2(us90)II* is aberrant.

The observed results could also be explained by an altered kinetics of the HSR activation in the *ctl-2(ua90)II* mutant. We have, therefore, measured the transcript levels of HSP16.1, HSP16.2, and HSP70 every hour for a total of 4 h of heat shock (Fig S1). For each measured transcript, a gradual increase in their level was observed, approaching saturation already after 2 h of heat shock for the WT and the *ctl-2(ua90)II* mutant (Fig S1). This result showed that the kinetics of the HSR are similar between the WT and *ctl-2(ua90)II* mutant and that the differences in the transcript levels after 4 h of heat shock are due to aberrant HSR in the *ctl-2(ua90)II* mutant.

The up-regulation of HSPs during HS is coordinated by the transcription factor HSF-1, which is kept inactive at OGT by binding to chaperones. When the levels of unfolded proteins increase, the chaperones are recruited to these proteins, releasing HSF-1, which forms a homotrimer that binds to sequences known as heat shock elements at the promoters of its target genes, from where it stimulates transcription. Formation of nuclear foci of the GFP-tagged HSF-1 has been shown to be induced by heat stress and

to colocalize with markers of active transcription (Morton & Lamitina, 2013). Quantification of the HSF-1::GFP foci showed fewer foci in the *ctl-2(ua90)II* worms compared with WT (Fig 1F and G), suggesting altered regulation of the HSR in the peroxisomal mutant. There are several possibilities here: formation of the "active" HSF-1 homotrimer may be reduced because of increased HSP-70 concentration or by another mechanism, HSF-1 DNA binding may be decreased or shortened, or the *ctl-2(ua90)II* strain may express less HSF-1 overall. Further experiments are needed to determine the exact mechanisms of the changes in HSR regulation in *ctl-2(ua90)II* strain, but it is clear that HS affects WT and *ctl-2(ua90)II* differently. The decreased activation of HSR and lower up-regulation of sHSPs is also supported by decreased thermotolerance of the *ctl-2(us90)II* strain. Survival measurement at the restrictive temperature of 37°C showed that WT worms were more heat tolerant compared with *ctl-2(ua90)II*; at 37°C, all *ctl-2(us90)II* worms were dead after 5 h compared with 8 h for the WT strain (≈40% difference) (Fig 1H).

### Aberrant stress responses in the peroxisomal mutants

HS causes a range of changes inside the cell, from misfolding proteins to metabolic changes. Therefore, we explored in more detail the expression levels of target genes from both aspects. We found that the unfolded protein response (ER UPR) was activated during HS in WT but was impaired in the *ctl-2(ua90)II* mutant (Fig 2A). Although in the WT, the expression of the UPR markers UGGT-1, PDI-6, and CRT-1 (Park et al, 2001; Buzzi et al, 2011; Eletto et al, 2014) was strongly increased (~1.8-, 5-, and 4-fold, respectively), in the *ctl-2(ua90)II* mutant, the increase was less prominent (~1.1-, 3-, and 1.8-fold, respectively). Mitochondrial UPR (mt UPR), on the other hand, was not activated in *ctl-2(ua90)II* strain and in WT was evidenced only by the up-regulation of peptidase CLPP-1 (Fig 2B) (Haynes et al, 2007). It was reported previously by Ben Zvi and colleagues (Ben-Zvi et al, 2009) that HSR and ER UPR activation is constrained in older animals, which would support the idea that *ctl-2(ua90)II* strain exhibits a progeric-like phenotype (Petriv & Rachubinski, 2004).

We have previously demonstrated that budding yeast experiences oxidative stress during HS (Musa et al, 2018), which also appeared to be the case in WT *C. elegans*, as suggested by multiple parameters measured herein, including the increase in transcript levels of SOD enzymes (Fig 2C). qRT-PCR measurement of superoxide dismutase (SOD) enzymes revealed that both cytosolic (SOD-1 and SOD-5) and mitochondrial (SOD-2 and SOD-3) superoxide dismutases were up-regulated in the WT during HS (~15-, 30-, 15-, and 110-fold), whereas HS did not affect transcript levels of these enzymes in the *ctl-2(ua90)II* strain. However, transcript levels of the cytosolic SOD-5 in *ctl-2(ua90)II* strain were increased ~15-fold already at OGT compared with WT and were comparable to WT HS levels (Fig 2C). The expression level of the SOD-5 remained unchanged in the *ctl-2(ua90)II* strain at HS.

Pentose phosphate pathway (PPP) is an associated biosynthetic pathway that also contributes, via NADPH production, to the response against oxidative stress but can also be induced during heat stress as a consequence of increased superoxide production (Stincone et al, 2015; Musa et al, 2018). Although transcript levels of PPP rate–limiting enzyme glucose-6-phosphate 1-dehydrogenase

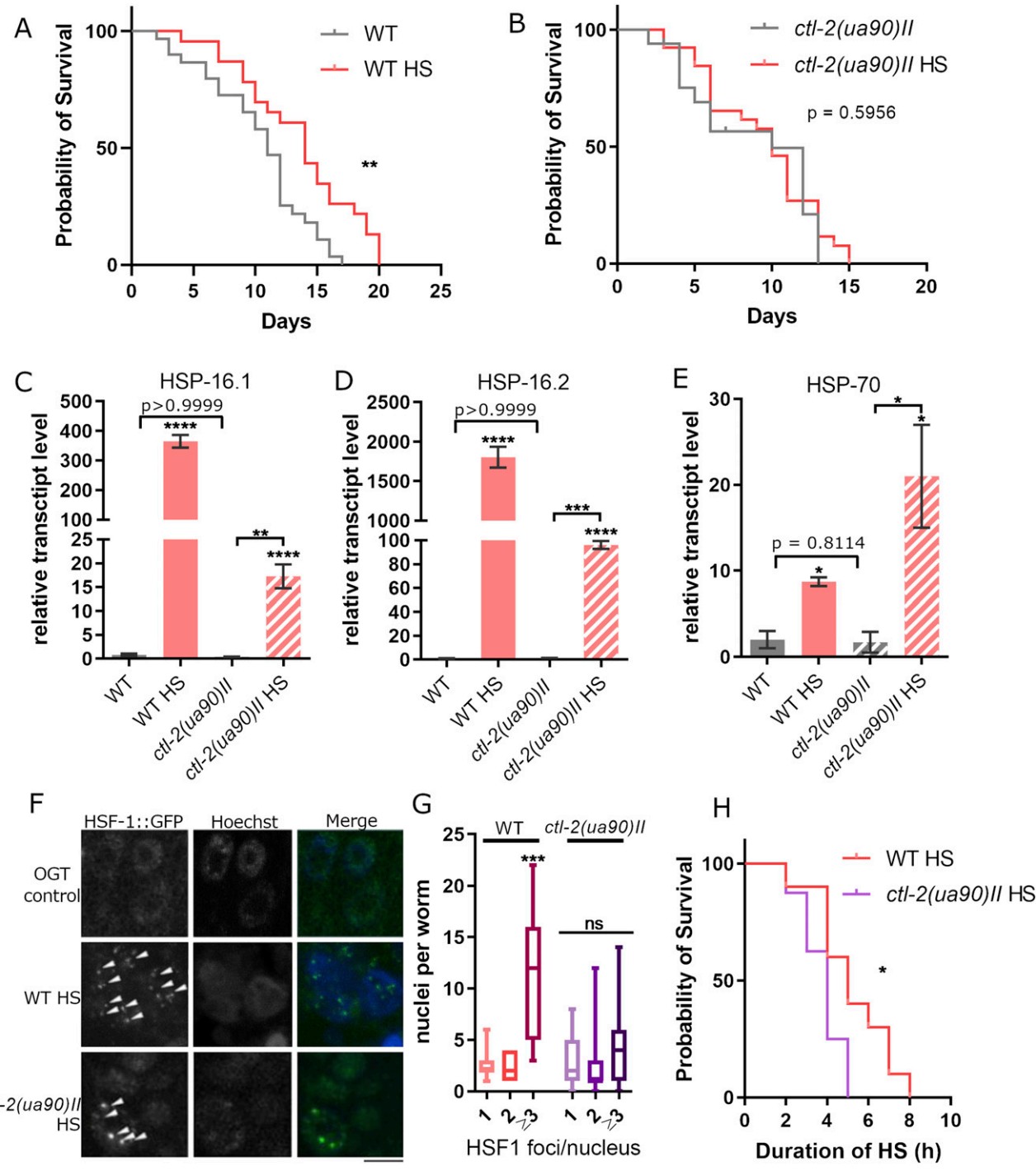

**Figure 1. Heat shock response is constrained in peroxisomal *ctl-2(ua90)II* mutant.**

**(A)** Mild transient HS at L4 stage extends WT lifespan. Median lifespan after HS was 14 d compared with 11 d at optimal growth temperature, and maximum lifespan was extended from 17 to 20 d after HS. *P* = 0.0048 (Mantel–Cox). **(B)** In *ctl-2(ua90)II*, background lifespan was not affected by HS; median lifespan was 10 d for both conditions; and maximum lifespan was 13 d at optimal growth temperature and 15 d after HS treatment. *P* = 0.5956 (Mantel–Cox) HS was administered at day 0. The curves are representative of three biological replicates; at least 100 worms were counted in each replicate. **(C, D, E)** Heat shock response was activated in both strains after 4-h HS; however, transcript levels of HSP16.1 and HSP16.2 were lower in *ctl-2(ua90)II* compared with WT. HSP-70 expression was higher in *ctl-2(ua90)II* mutant compared with WT after HS. Gene expression normalized to *act-1*. Error bars are mean of biological and technical duplicates ± SD. ****P < 0.0001; ***P < 0.001; **P < 0.01; *P < 0.05 (one-way ANOVA plus Tukey test). **(F)** HSF-1::GFP forms foci in the nucleus during HS in the WT. The black bar represents 5 μm. **(F, G)** Quantification of images displayed in (F) shows that *ctl-2(ua90)II* mutants have a significantly lower proportion of nuclei with more than three foci: WT averages 15 nuclei with three or more HSF1 puncta after 4 h of HS compared with only four nuclei in *ctl-2(ua90)II*. Nuclei were quantified in the intestine of live worms; experiment was repeated twice. ***P < 0.001; **P < 0.01; *P < 0.05 (one-way ANOVA plus Tukey test). **(H)** *ctl-2(ua90)II* mutant shows decreased ability to withstand heat stress. The curves represent biological duplicates and a minimum of 100 worms for each experiment. *P* = 0.037 (Mantel–Cox).

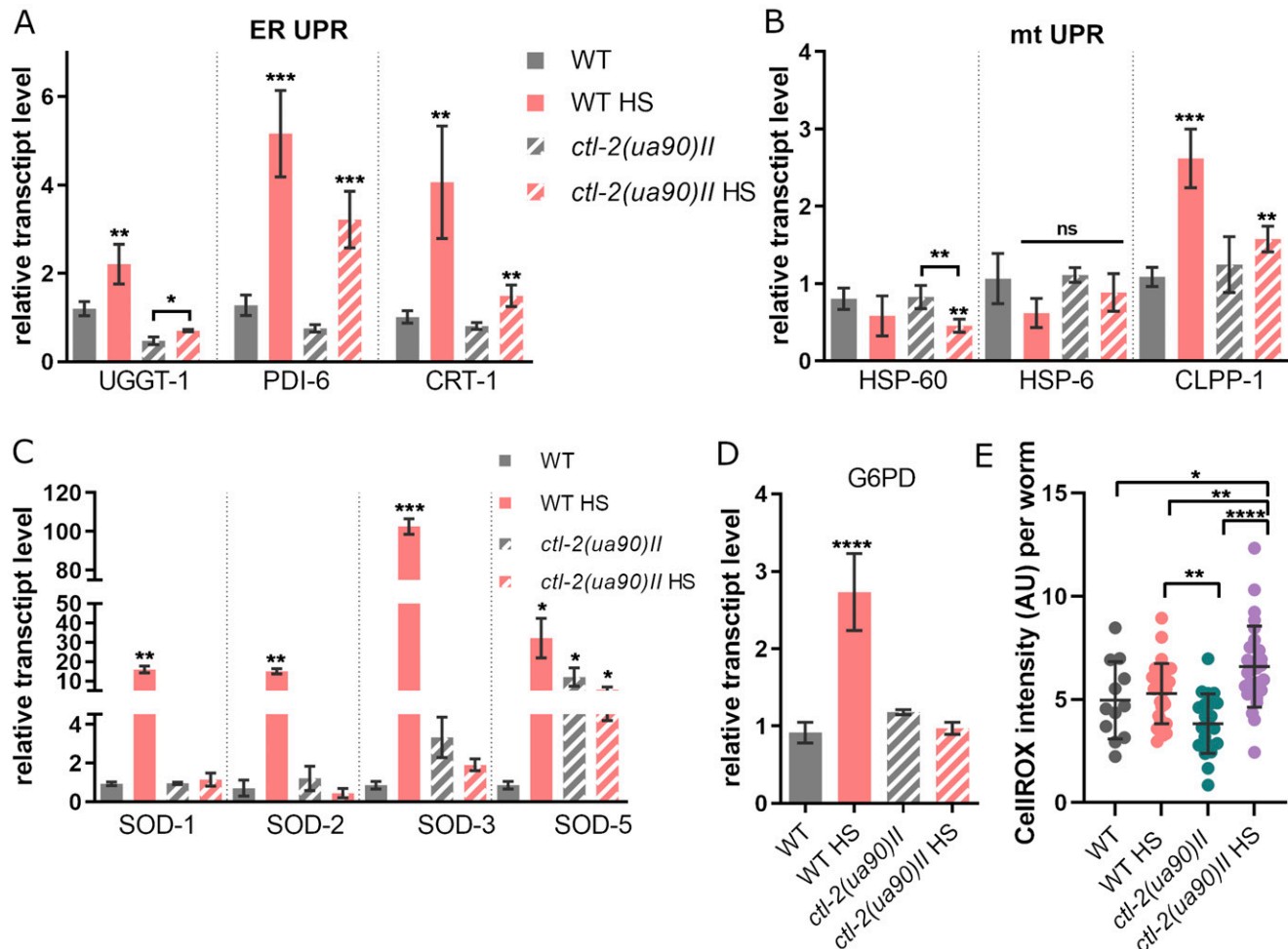

**Figure 2. *ctl-2(ua90)II* strain reacts differently to HS compared with WT outside of heat shock response as well.**
**(A)** ER UPR was activated in both WT and *ctl-2(ua90)II* strains. **(B)** mtUPR was modestly up-regulated in WT, as evidenced by increase in expression of CLPP-1 caspase but was absent in *ctl-2(ua90)II* strain. **(C)** HS does not affect transcript levels of SOD enzymes in the *ctl-2(ua90)II* strain. **(D)** HS increases transcript levels of G6PD, rate-limiting enzyme of pentose phosphate pathway in WT but not in *ctl-2(ua90)II* strain. Gene expression normalized to *act-1*. Error bars are mean ± SD for biological replicates. **(E)** ROS levels are comparable between WT and *ctl-2(ua90)II* mutant but are increased in *ctl-2(ua90)II* during HS. ****$P < 0.0001$; ***$P < 0.001$; **$P < 0.01$; *$P < 0.05$ (one-way ANOVA plus Tukey test).

(GSPD-1) as measured by qRT-PCR were expectedly increased (~2.8-fold) in WT during HS (Stincone et al, 2015; Musa et al, 2018), this was not the case in *ctl-2(ua90)II* mutant, suggesting that *ctl-2(ua90)II* strain did not activate PPP during HS (Fig 2D). Lack of up-regulation of key antioxidative enzymes during HS indicates that *ctl-2(ua90)II* strain does not fully sense oxidative stress during HS, presumably as a consequence of catalase deficiency, or is unable to respond properly. As SOD-5 seems to be constitutively up-regulated in *ctl-2(ua90)II* strain (Fig 2C), it is possible that lack of further up-regulation of chaperones is an adaptation to avoid toxic effects of prolonged activation of stress response pathways and that the threshold for their activation is increased in this strain.

To estimate the level of ROS in *C. elegans*, we used CellROX dye-based assay, whose fluorescence is proportional to the amount of ROS in the cell. Interestingly, the CellROX fluorescence in WT was not affected by HS, likely as a result of the activation of antioxidant defenses. In the *ctl-2(ua90)II* strain, the ROS levels were comparable to the WT strain at OGT, as indicated by CellROX, but were

significantly increased after HS (~1.75-fold increase in median fluorescence compared with the OGT) (Fig 2E). Therefore, it is plausible that the high levels of SOD enzymes in the WT are efficiently containing the ROS levels and preventing their accumulation during HS, whereas the increasing ROS levels in the *ctl-2(ua90) II* strain during HS are not neutralized. The mechanisms preventing the *ctl-2(ua90)II* mutant from activating the antioxidant defenses remain to be elucidated.

## Peroxisome number and morphology are affected by HS

Peroxisomes are known to change in shape, size, and abundance in response to environmental stimuli and cell status. Given the differences observed in the response to HS between the WT and the peroxisomal mutant *ctl-2(ua90)II*, we turned our attention to the peroxisome morphology and function. In this context, the peroxisome import machinery is important for proper assembly of the organelle. We measured the expression levels of two peroxisomal

transport proteins, PRX-5 and PRX-11, and found no differences in expression between WT and *ctl-2(ua90)II* strains (Fig 3A).

Peroxisomes were visualized using GFP fused with peroxisomal import signal (PTS::GFP; Fig 3B). Already at OGT, the peroxisomes in the *ctl-2(ua90)II* strain were less abundant, or less able to import the proteins, than in the WT (Fig 3C). Moreover, imaging revealed decreased number of peroxisomes after HS in both WT and *ctl-2(ua90)II* strain compared with OGT of each strain (Fig 3C). *ctl-2(ua90)II* peroxisomes were also characterized by a slightly larger

size compared with WT at OGT, measured as peroxisome area in square pixels (Fig 3D), consistent with previous reports (Petriv & Rachubinski, 2004). Interestingly, although WT peroxisomes tended to increase in size during HS, the size of *ctl-2(ua90)II* peroxisomes decreased (Fig 3D).

While imaging peroxisomes, we also observed subtle changes in their shape between WT and *ctl-2(ua90)II* strains. To quantify changes in the peroxisome shape between the strains, we measured them along the shortest and longest axes, the ratio of which

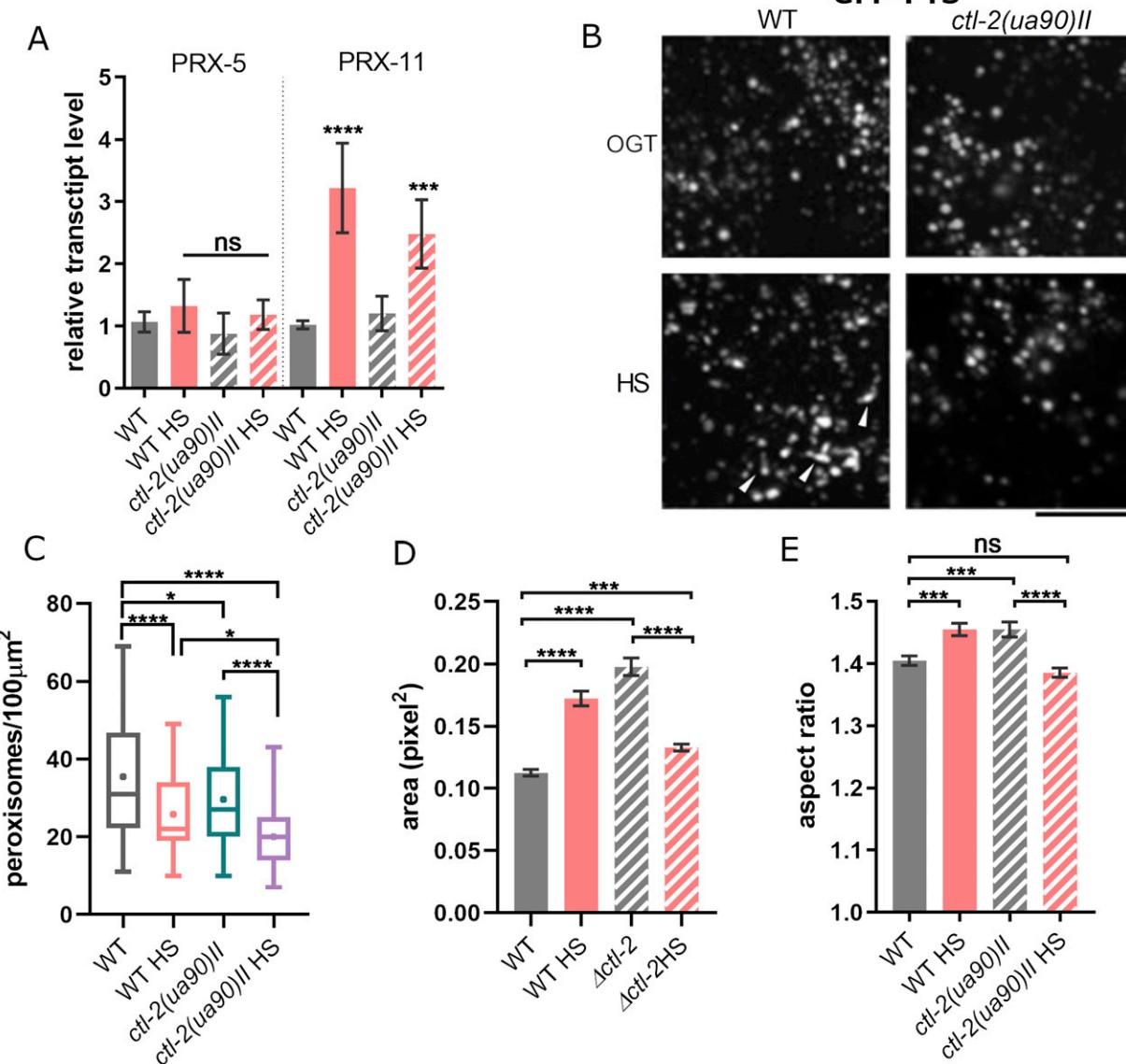

**Figure 3. Peroxisome morphology and number are affected by mild transient HS.**
**(A)** Transcript levels of main biogenesis proteins in peroxisomes responsible for transporting its contents are similar in WT and *ctl-2(ua90)II* both in control and HS-treated samples, suggesting there is no significant change in peroxisomal import in *ctl-2(ua90)II* strain compared with WT. Gene expression normalized to *act-1*. Error bars are mean ± SD for biological replicates. ****$P < 0.0001$; ***$P < 0.001$; **$P < 0.01$; *$P < 0.05$ (ANOVA plus post hoc). **(B)** Peroxisomes were visualized with a GFP-PTS fusion protein. The black bar represents 5 μm. **(B, C)** Quantification of the images represented in (B) shows that, on average, WT worms have more peroxisomes than *ctl-2(ua90)II* strain. Their number is further reduced during HS. The symbol signifies the mean. **(D)** *ctl-2(ua90)II* peroxisomes are on average larger than those in WT; during HS, WT peroxisomes increase in size, whereas *ctl-2(ua90)II* peroxisomes become smaller. **(E)** Aspect ratio of the imaged peroxisomes shows that WT peroxisomes become more elongated during HS, whereas in *ctl-2(ua90)II*, the trend is opposite. The graphs represent mean ± SEM of 10 worms for each condition from two experiments. ****$P < 0.0001$; ***$P < 0.001$; **$P < 0.01$; *$P < 0.05$ (one-way ANOVA plus Tukey test).

approximates their sphericity: aspect ratio of 1 would describe a perfect sphere, whereas any increasing values indicate elongation. At OGT, the WT peroxisomes appear slightly more spherical than *ctl-2(ua90)II* peroxisomes with ~4.5% increase in the aspect ratio. During HS, the sphericity of the WT peroxisomes was reduced with the aspect ratio increase of ~5%, suggesting elongation. On the other hand, the *ctl-2(ua90)II* peroxisomes displayed a tendency to become more spherical during HS (Fig 3E), with aspect ratio decrease of ~6% compared with the same strain at OGT. More experiments are necessary to dissect the causes and importance of these changes. The observed changes are subtle but likely indicate changes in peroxisomal maintenance between the two strains.

### Fatty acid metabolism, synthesis, and storage are affected by HS and ctl-2(ua90)II mutation

One of the most important processes taking place in the peroxisomes is oxidation of very long fatty acids, products of which are stored as a reserve energy supply or shuttled to mitochondria to be fully oxidized and used to produce ATP. Although mitochondria can oxidize most FAs themselves, very long chain FAs can only be oxidized in the peroxisomes. For example, phytanic acid is converted to pristanic acid via α-oxidation, then broken down further through multiple rounds of β-oxidation in the peroxisome to FAs that can be fully oxidized in mitochondria. Therefore, changes in the peroxisome metabolism could cause changes in substrate availability for the mitochondrial β-oxidation. Curiously, we found that peroxisomal β-oxidation is suppressed during HS in both WT and *ctl-2(ua90)II*, evidenced by down-regulation of gene encoding peroxisomal straight-chain acyl-CoA oxidase, ACOX-1, the rate-limiting enzyme for very long chain FA β-oxidation (Fig 4A). After HS, both WT and *ctl-2(ua90)II* strains displayed ~25% of the transcript levels measured at OGT (Fig 4A). A similar trend was observed in the second step of peroxisomal β-oxidation, catalysed by MAO-C-like dehydratase domain protein MAOC-1 (Fig 4B). However, the last step, the thiolytic cleavage of 3-ketoacyl-CoA catalysed by DAF-22 (propanoyl-CoA C-acyltransferase), was unaffected by HS in WT. Interestingly, in *ctl-2(ua90)II* strain, the transcript levels for enzyme DAF-22 were increased twofold at OGT compared with WT and decreased significantly after HS (Fig 4C). These results may suggest that this aspect of peroxisomal function may be important in the context of response to HS. The differences in the transcript levels of β-oxidation enzymes between WT and *ctl-2(ua90)II* strain leave open a possibility that the levels of β-oxidation intermediates and final products are changed in the *ctl-2(ua90)II* mutant compared with WT, thus possibly affecting both mitochondrial function and the energy supply of the cells in the form of fatty acids.

In addition, we measured the transcript levels of several peroxisomal enzymes to characterize the function of peroxisomes. Lon protease homolog, lonp-2, displayed an increase in the expression level during HS in the WT and the *ctl-2(ua90)II* worms (Fig S2). We measured the transcript levels of a peroxisomal enzyme with oxidoreductase activity, F41E6.5, and carnitine O-octanoyltransferase, T20B3.1; however, they did not change during HS or in the mutant (Fig S2).

Proper metabolism of FAs is essential for life, especially in the context of HS, as lipids are major components of the cellular membrane. The membrane alters its composition to adapt to environmental conditions and deal with different stressors, such as temperature. Because our results suggest that FA β-oxidation is affected by HS, we further aimed to assess FA synthesis and storage. To that end, we measured the transcript levels of major FA synthesis enzymes and visualized FA stores in fixed worms. We found distinct differences in the transcript levels of the rate-limiting enzyme acetyl-CoA carboxylase (POD-2), which was roughly tripled at HS in both strains, but the levels of POD-2 coding transcripts were also lower at OGT in *ctl-2(ua90)II* compared with WT (Fig 4D). Although there were no significant changes in transcript levels of fatty acid synthase (FASN-1) between WT and *ctl-2(ua90)II* (Figure 4E), Δ9 desaturases FAT-5 and FAT-6 were differently regulated. More specifically, FAT-5 expression was roughly doubled after HS in WT, whereas in the *ctl-2(ua90)II* mutant, it was halved (Fig 4F). Furthermore, the FAT-6 expression was unchanged after HS in WT, whereas in the *ctl-2(ua90)II* mutant, it was increased by ~25% (Fig 4G). As FAT-5 and FAT-6 have different substrate specificities (FAT-5 readily desaturates palmitic acid [16:0] but not stearic acid [18:0] which is desaturated by FAT-6), the differences in their expression between strains indicate that the proportions of different FA species in the two strains could be impacted. As lipid composition is crucial, especially in conditions of changing temperature, these changes could have major impact on the worm's survival during HS.

FAs are stored mainly in the *C. elegans* intestine inside so-called lipid droplets. Lipid droplets can be visualized using Nile red dye, which stains neutral lipids (Fig 4H). Nile red staining revealed no significant difference between strains nor any discernible effect of the HS treatment (Fig 4I–K). However, Nile red does not differentiate between different lipid species. Therefore, we stained the worms using Oil Red O dye which stains triglycerides (Fig 5A and B). Oil Red O was not significantly affected by HS, but it revealed a slight difference between the two strains, suggesting that triglyceride production and/or storage is affected in the *ctl-2(ua90)II* worms compared with WT as they retained less of the Oil Red O dye, indicating lesser triglyceride content in the *ctl-2(ua90)II* mutant (Fig 5C–K). This is consistent with the qRT-PCR results revealing differences in FA synthesis between the two strains as decreased transcription of POD-2 is usually associated with decreased triglyceride levels (Kim et al, 2016). Although further experiments are needed to investigate the details of the lipid production, storage, and usage in the *ctl-2(ua90)II* strain, these data provide evidence that there are differences in the FA content between the peroxisomal mutant and WT.

### Mitochondrial morphology is altered in ctl-2(ua90)II mutant during HS

Given the metabolic connections between peroxisomes and mitochondria and the changes we observed in peroxisomes, we turned to mitochondria next. Mitochondria form physical contact sites with peroxisomes to communicate and facilitate exchange of molecules such as lipids from peroxisomes that are to be used by mitochondria to produce ATP. Mitochondria were visualized with MitoTracker Deep Red dye and imaged in live worms using spinning disc confocal microscope (Fig 6A). Analysis of stained mitochondria indicated that WT and *ctl-2(ua90)II* mitochondria displayed differences in mitochondrial morphology. The number of branches, branch junctions, and the average branch length were used to

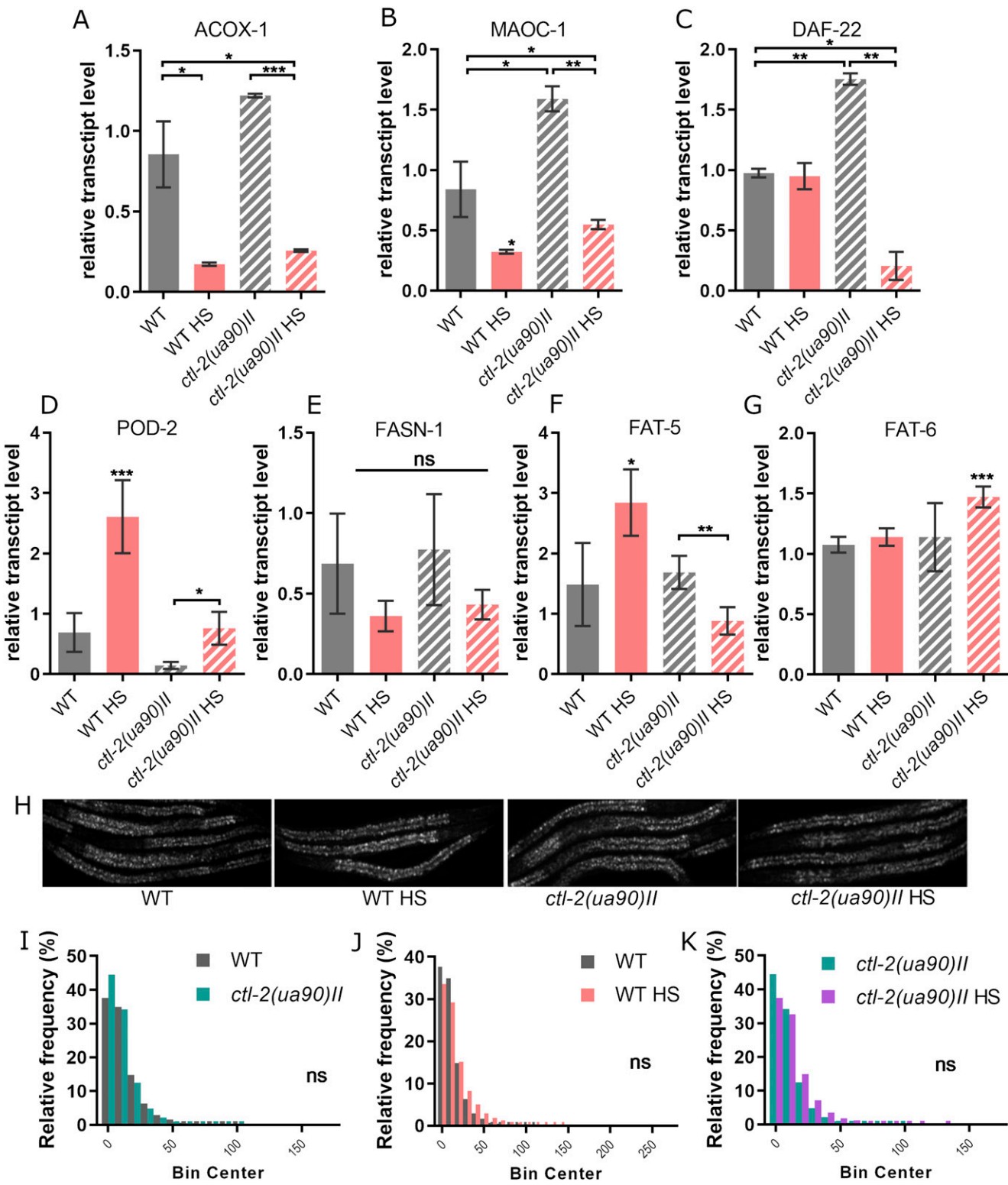

**Figure 4.   Fatty acid metabolism is altered in the *ctl-2(ua90)II* mutant.**
**(A, B, C)** Major peroxisomal FA β-oxidation enzymes (A) ACOX-1, (B) MAOC-1, and (C) DAF-22 expression levels are increased in the *ctl-2(ua90)II* mutant at optimal growth temperature compared with WT. **(D, E, F, G)** *ctl-2(ua90)II* mutant displays different regulation of genes involved in fatty acid synthesis during HS compared with WT. Gene expression normalized to *act-1*. Error bars are mean ± SD of biological replicates. ****P < 0.0001; ***P < 0.001; **P < 0.01; *P < 0.05 (one-way ANOVA plus Tukey test). **(H)** Nile red staining of lipid droplets. **(H, I, J, K)** The size of lipid droplets from (H) was not significantly different between WT and *ctl-2(ua90)II* mutant and was not affected by HS. Bin center represents lipid droplet area measured in pixels. At least 10 worms were quantified for each condition. Experiment was repeated three times. ****P < 0.0001; ***P < 0.001; **P < 0.01; *P < 0.05 (Mann–Whitney).

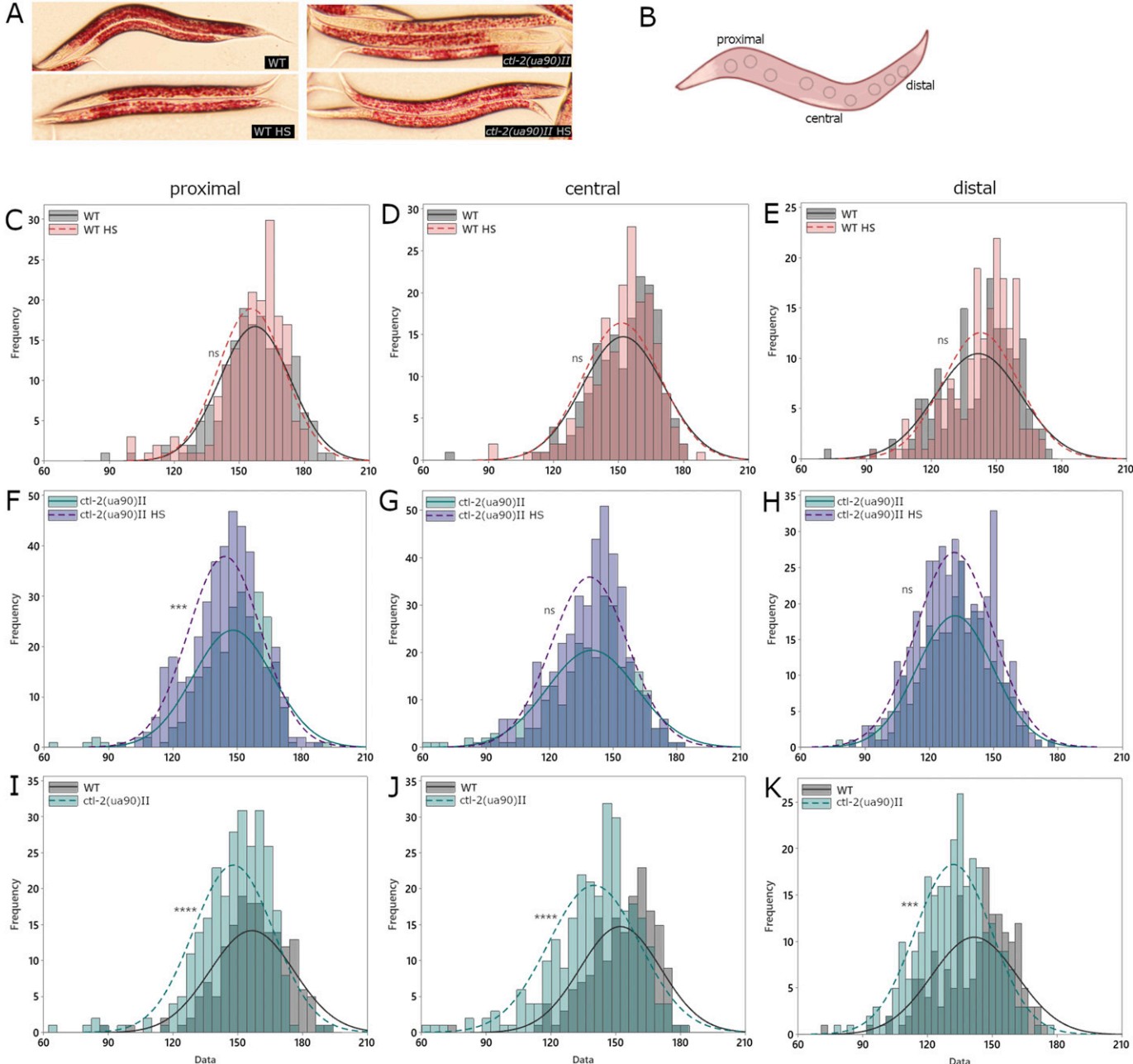

**Figure 5. Triglyceride levels are altered in *ctl-2(ua90)II* strain.**
**(A)** Raw images of Oil Red O–stained worms. **(B)** Staining intensity was measured separately in the area between the pharynx and the vulva (proximal), area around the vulva (central), and the tail section (distal), avoiding the darker edge of the worm and the vulva, as integrated density, that is, sum of values of pixels normalized to area. Three measurements of each section were averaged for each worm, at least 20 worms from two experiments for each condition. **(C, D, E, F, G, H, I, J, K)** *ctl-2(ua90)II* worms show lower intensity Oil Red O staining compared with WT, indicating lower triglyceride content. ****$P < 0.0001$; ***$P < 0.001$; **$P < 0.01$; *$P < 0.05$ (Mann–Whitney).

describe mitochondrial networks, whereas individual mitochondria were separated based on their shape. The analysis of the mitochondrial networks shows that the average *ctl-2(ua90)II* strain mitochondrion is less branched, with fewer, shorter branches compared with WT. WT mitochondrial branches seem to be ~25% longer than those observed in *ctl-2(ua90)II* strain at OGT. After HS, however, the average branch length is decreased by ~50% in WT, whereas it is increased by ~20% in *ctl-2(ua90)II* strain (Fig 6B). In both WT and *ctl-2(ua90)II* strains, mitochondria appear less branched after HS compared with OGT, indicated by the decrease in the number of branches and branch junctions (Fig 6C and D). Analysis of the non-branched mitochondria reveals further differences (Fig 6E–G). At OGT, long, rod-like mitochondria were most commonly observed. However, at HS, we observed a greater

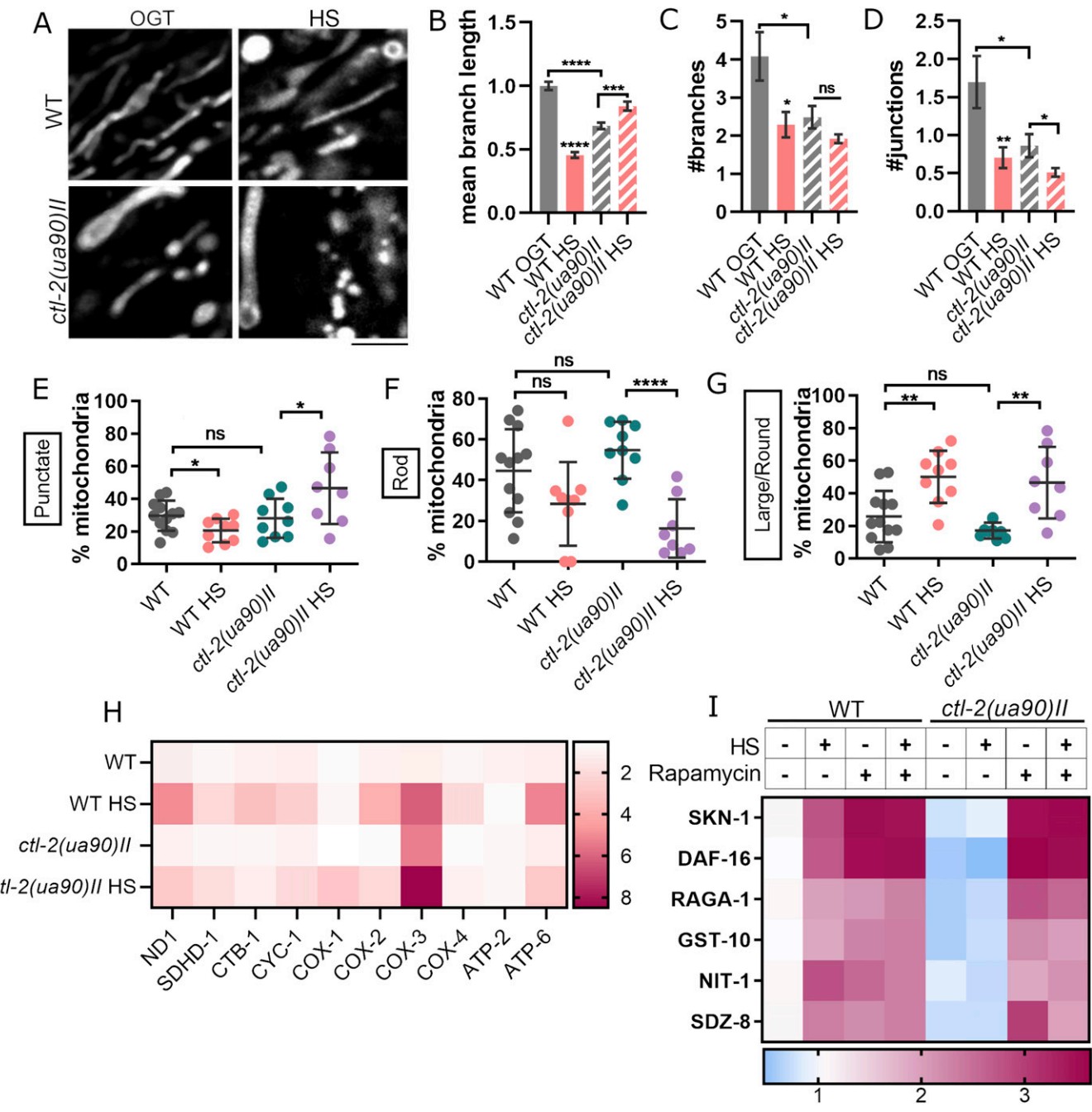

**Figure 6. Mitochondrial transcript levels and mitochondrial morphology are altered in the *ctl-2(ua90)II* strain.**
**(A)** Imaging of MitoTracker-stained mitochondria shows differences in mitochondrial morphology between WT and *ctl-2(ua90)II* mutant. **(B, C, D)** Analysis and quantification of networked mitochondria reveals that HS results in fewer and shorter branches, that is, less branched mitochondrial networks. *ctl-2(ua90)II* strain also has less branched mitochondria at optimal growth temperature. ****$P < 0.0001$; ***$P < 0.001$; **$P < 0.01$; *$P < 0.05$ (unpaired $t$ tests with Welch's correction). **(E, F, G)** Comparison of individual mitochondria in WT and *ctl-2(ua90)II* strain shows that non-branched mitochondria are similar at optimal growth temperature in both strains, but during HS, WT displays more large, globular mitochondria, whereas *ctl-2(ua90)II* mutant has more punctate mitochondria. **(H)** Respiratory chain subunits are increased during HS in both strains but to a lesser extent in the *ctl-2(ua90)II* strain, except for COX-4 subunit of complex IV. **(I)** RT–PCR measurement reveals increased transcript levels of TORC1 downstream targets, indicating TORC1 has been inhibited in the WT during HS but not in the peroxisomal mutant. Color of the squares on the heat map corresponds to the mean value of the log fold change. Transcript levels were normalized to *act-1*. ****$P < 0.0001$; ***$P < 0.001$; **$P < 0.01$; *$P < 0.05$ (two-way ANOVA with Geisser–Greenhouse correction).

proportion of large and round mitochondria in WT, whereas *ctl-2(ua90)II* strain accumulates both large and round mitochondria and small, punctate mitochondria. The small, punctate mitochondria are usually indicative of active mitochondrial fission.

We further measured transcript levels of respiratory chain subunits. During HS, respiratory chain subunit encoding genes were generally up-regulated, with a few exceptions; CTB-1 subunit of complex III was not up-regulated during HS in *ctl-2(ua90)II* strain,

whereas complex IV COX-1 (CTC-1) subunit was up-regulated only in the *ctl-2(ua90)II* mutant, ~3-fold compared with WT. Furthermore, transcript levels for complex IV subunit COX-3 (CTC-3) were ~6 times higher in WT during HS, which was comparable to the level measured in *ctl-2(ua90)II* strain both at OGT and HS (Figs 6H and S3). Although these results suggest changes in respiratory metabolism during HS, they do not point toward different trends between WT and *ctl-2(us90)II* strains.

### TORC1 is inhibited in WT during HS but not in the ctl-2(ua90)II strain

Finally, we measured TORC1 activity during HS. TORC1 inhibition has been repeatedly demonstrated to extend lifespan across species (Bonawitz et al, 2007; Pan & Shadel, 2009; Wei et al, 2009; Pan et al, 2011; Robida-Stubbs et al, 2012; Johnson et al, 2013; Semchyshyn & Valishkevych, 2016; Perić et al, 2017; Musa et al, 2018). Furthermore, we have previously found that TORC1 is inhibited after mild HS in budding yeast (Musa et al, 2018). In *C. elegans*, TORC-1 has been associated with HSR through its interaction with homeodomain-interacting protein HPK-1 (Das et al, 2017). When TORC1 is inhibited, its downstream target SKN-1/Nrf is uninhibited and activates transcription of protective genes, such as superoxide dismutases and catalases, and TORC1 subunits as a part of a feedback loop. DAF16/FoxO is also activated upon TORC-1 inhibition: both DAF-16/FoxO and SKN-1/Nrf are inhibited by insulin/IGF-like signaling in *C. elegans* (Kenyon, 2010; Kenyon, 2011; Robida-Stubbs et al, 2012). As we did not have access to a reliable phospho-S6 antibody, we measured TORC-1 activity indirectly by qRT-PCR measurement of its downstream targets: SKN-1, DAF-16, RAGA-1, GST-10, NIT-1, SDZ-8. The qRT-PCR measurement suggested that TORC-1 was inhibited by both HS and rapamycin in the WT strain, indicated by increased transcription of TORC-1 downstream targets, including TORC-1 subunit RAGA-1. Treatment with rapamycin, inhibitor of TORC-1, triggered a similar increase in TORC-1 downstream targets, suggesting that, like rapamycin, HS leads to TORC-1 inhibition. Interestingly, HS did not inhibit TORC-1 in the peroxisomal *ctl-2(ua90)II* mutant, whereas rapamycin did (Figs 6I and S4). Hence, the TORC-1 activity in the *ctl-2(ua90)II* mutant may be contributing to its shortened lifespan and lack of hormetic effect after mild HS.

## Discussion

Hormetic effects after exposure to low doses of stressors have been observed to date in several organisms (Butov et al, 2001; Cypser & Johnson, 2002; Hercus et al, 2003; Calabrese et al, 2007; Rattan & Demirovic, 2009; Semchyshyn & Valishkevych, 2016; Kumsta et al, 2017). However, the general mechanisms of how these effects arise are still a matter of debate. What is widely accepted is that cells up-regulate various cellular defense mechanisms such as chaperones in response to the appearance of unfolded proteins during heat stress and that this increase in chaperone concentration and improvement of folding conditions is a crucial component of hormetic lifespan extension (Richter et al, 2010; Morimoto, 2011). However, several lines of evidence exist that suggest HSR, although

necessary, is not by itself sufficient to support stress-induced lifespan extension. In yeast, for example, it has been demonstrated that, for the mother cell to benefit from heat shock, the diffusion barrier between the mother and the nascent daughter cell needs to be permeated, allowing for diffusion of damaged cell components and aging factors, which would otherwise be retained by the mother, into the daughter cell (Baldi et al, 2017). Moreover, mitochondrial superoxide has been shown to be required for stress-induced lifespan extension in both yeast and *C. elegans* (Van Raamsdonk & Hekimi, 2009; Musa et al, 2018).

Furthermore, budding yeast strains that were not able to switch from fermentative to respiratory growth or increase respiration rate during HS did not display extended lifespan despite up-regulation of HSPs (Perić et al, 2017; Musa et al, 2018). Even HSR itself, which is traditionally discussed in terms of HSF-1, HSP-70, and HSP-90 in a regulatory loop, has been shown to be a part of a more vast and complicated regulatory network that is needed to regulate the HSR in different tissues (Guisbert et al, 2013). As a part of this wider regulatory network, homeodomain-interacting protein HPK-1 has been found to act upstream of HSF-1 and to be required for HSF-1-controlled chaperone up-regulation and TORC-1 inactivation (Das et al, 2017). These observations demonstrate that any hormetic effect must be a consequence of proper management and fine-tuning of cellular processes during stress, including canonical stress response pathways, as opposed to increased concentration of chaperones alone or activation of a single pathway alone.

The results presented here further support this idea. Unlike plant peroxisomes, animal peroxisomes have not been ascribed any vital roles in the HSR so far. However, the lack of peroxisomal catalase seems to have impaired the *ctl-2(ua90)II* mutant's ability to properly respond to heat stress, evidenced by lack of lifespan extension, decreased thermotolerance, and a range of metabolic changes when compared with WT worms, which are not confined to peroxisomes. Although we do not know the exact place peroxisomes take in the cascade of cellular events from the onset of HS to lifespan extension, it seems that fully functional peroxisomes are necessary for the hormetic effects to occur. Several possibilities can be speculated about based on the presented results. For example, in the absence of the peroxisomal catalase, it is likely that $H_2O_2$ accumulated in the peroxisomal matrix triggers the Fenton reaction when in contact with divalent iron. In this way, peroxisomal machineries may suffer oxidative damage, leading to their compromised function. Furthermore, changes in the FA metabolism may consequently negatively impact membrane composition, making it more prone to damage by oxidation or affecting its fluidity. As peroxisomes are sites of lipid catabolism, providing substrates such as shortened FAs for mitochondria to be used to produce energy, disrupting their function may in turn disrupt mitochondrial metabolism as well, causing a chain reaction of dysregulation. For example, the observed decrease in transcription of the rate-limiting enzyme POD-2 could result in a deficit of malonyl CoA and inhibition of transport of FAs to mitochondria (Bowman & Wolfgang, 2019). Decreased peroxisomal β-oxidation was also shown to increase mortality in yeast on rich media as a result of buildup of neutral FAs and TAGs (Titorenko & Terlecky, 2011). Furthermore, FA metabolism and lipid droplet dynamics have both been implicated in regulating longevity through modulation of availability of certain FAs and

lipophilic hormones such as dafachronic acid and pregnenolone which delay aging in *C. elegans* (Russell & Kahn, 2007). Although we did not observe buildup of FAs and TAGs in *C. elegans*, we cannot exclude the possibility that some lipid species were increased in abundance at the expense of others after either HS or in the absence of the peroxisomal catalase based on the metabolic changes we observed. However, more precise detection methods would be required to observe these changes.

We also observed minor but potentially biologically relevant changes in the peroxisome morphology. As peroxisomes are known to be plastic organelles that change in response to changing cellular and environmental conditions, the subtle changes we observed even at OGT might point to more widespread changes in how the peroxisomal catalase mutants manage their peroxisomes and their mitochondria. As there is no active peroxisomal peroxidase in the *ctl-2(u90)II* mutants, the peroxide produced by the FA oxidation in the peroxisomes may be damaging the peroxisomes themselves but also possibly leaking out of the peroxisome and damaging the cytoplasmic proteins before it can be neutralized by the cytosolic peroxidase. Paired with the changes in the peroxisomal metabolism of FAs and the potentially changed FA makeup of the *ctl-2(ua90)II* cells, it would be interesting to explore if these cells are more prone to oxidation and to what extent.

Finally, the lack of TORC-1 inactivation alongside decreased HSR activation in the peroxisomal mutant highlight the complexity and intersectionality of the cellular stress response. TORC-1 activity has long been correlated with decreased lifespan across organisms (Zoncu et al, 2011; Robida-Stubbs et al, 2012; Johnson et al, 2013; Albert & Hall, 2015). In *C. elegans*, TORC-1 has been found to inhibit HPK-1, which acts upstream of HSF-1 and is suggested to be a part of the wider HSF-1 network (Das et al, 2017). Moreover, it was also previously described that sHsps expression, including sHsp16.1, is reduced in animals with decreased DAF-16 activity (Hsu et al, 2003). These findings provide some context for our observation of decreased chaperone expression paired with TORC-1 activity in catalase deficient worms, albeit the exact role of the peroxisomes, specifically peroxisomal catalase, is yet to be determined.

Together, these data support the notion that stress-induced lifespan extension is likely a result of a cell-wide coordinated response, alongside robustly activated canonical HSR. Interestingly, although differences between species do exist in how they respond to and benefit from stress, TORC-1 suppression, chaperone upregulation, and tightly regulated, localized increases of ROS seem to be the conserved mechanisms by which eukaryotic cells build up and maintain resistance to stress and prolong their lifespan. Animal peroxisomes have not been implicated in HSR, but given their roles in both ROS and FA metabolism, it is very likely that they may play a role in the stress response as well. Our results, although at this stage lacking detailed mechanistic insight, suggest that peroxisomes indeed have a larger role in the cellular stress response than ascribed thus far and would make a promising focus of study in this context in the future.

## Conclusion

Hormetic effects of mild, transient heat shock have been described for a range of organisms from yeast to mammals. Our goal was to explore the effects of HS which contribute to the hormetic effect but which are outside of the canonic HSR. Although further experiments are necessary to further define the exact role and function of peroxisomes in the context of hormesis, our results show that peroxisomes may have a more important role than previously ascribed and could possibly make an interesting new subject in hormesis and stress response research. Peroxisomes have a number of vital roles which may affect survival during stressful conditions, including but not limited to their connection with mitochondria which have already been implicated in many aspects of stress biology and their role in lipid and ROS metabolism; peroxisomes therefore may hold answers to many yet unanswered questions related to hormesis and stress-induced lifespan extension and possibly stress perception and cellular response to stress. Finally, better understanding of its role, and the more complete picture of heat stress response across the cell and outside of the canonical HSF-1 pathway, is key to applying the principles of lifespan extension we have learned from unicellular eukaryotes to better understand how different stresses are sensed and handled by more complex organisms.

## Materials and Methods

### Strains

*C. elegans* strains were obtained from Caenorhabditis Genetics Center at the University of Minnesota (https://cgc.umn.edu/). Animals were maintained at 20°C on NGM media seeded with OP50 *Escherichia coli*. To obtain a *ctl-2* mutant with GFP reporters, we crossed GFP-carrying strains (OG497 and VS15) as follows: unsynchronized populations abundant in L4 stage worms of GFP-carrying strains were repeatedly heat shocked at 30°C or 37°C for multiple generations until male worms were observed on the plate. The males were then picked and placed on a separate 3-cm plate seeded with OP50 with 2 or 3 L4 hermaphrodites to enrich for males. GFP-carrying males were then picked on separate 0P50-seeded plates together with *ctl-2(ua90)II* L4 hermaphrodites. Once the F2 worms reached L4 stage, individual F2 worms were picked onto individual plates to lay eggs, after which they were collected for screening under a fluorescent microscope. The eggs and the progeny of the worms which exhibited appropriate fluorescence were confirmed to be *ctl-2(ua90)II* using PCR (ctl2 Fw TTAGA-TATGAGAGCGAGCCTGTTTC; ctl-2 Rc CTAGTGGTACATCCATGCAAATGC).

### Synchronization

Mixed plates of worms were chunked onto fresh NGM plates seeded with OP50 and grown until they contained predominantly gravid adults. Gravid adults were washed from the plate with M9 buffer or water, and the plate was gently scraped with a spatula to collect the already laid eggs. The eggs and gravid worms were settled by gravity or brief centrifugation at low RPM in 15-ml Falcon tubes. The worm and egg pellet were incubated in 20% hypochlorite solution for 5 min with occasional vortexing, until adult hermaphrodites appear broken. The eggs are then settled by centrifugation and washed

with sterile M9 buffer two times. The pellet was then either incubated in a sterile 7-cm plate in 7 ml of sterile M9 buffer at RT with light shaking or seeded on a clean NGM plate where they can hatch without food. After hatching, the arrested L1 larvae are transferred to NGM plates seeded with appropriate food source and incubated at 20°C.

### Lifespan measurement

Worms were synchronized using the bleaching method and transferred to fresh OP50 NGM plates, 10–15 worms per 13-cm plate to avoid overcrowding. Worms were counted and transferred to a fresh plate every 2 d until they stopped moving or responding to prodding with the pick. A minimum of 100 worms were picked for each condition for every repetition to ensure sufficient number after censoring stragglers, bagged worms, and worms killed during transfer. Worms were kept at 20°C (OGT) continuously.

### Heat shock

Heat shock was performed on plates at 30°C. Worms were incubated at 20°C (OGT) until L4 stage when they were transferred to a 30°C incubator for HS for 1, 2, 3, or 4 h. After HS, they were either immediately collected for RNA isolation or staining without recovery or returned to 20°C for lifespan analysis.

### Thermotolerance assay

Plates with synchronized and well-fed L4 worms were taken from 20°C and incubated at 37°C for a set amount of time, after which they were left to recover at RT overnight. Survival was scored the next day; worms which were not moving and did not respond to stimuli were scored as dead. A minimum of 100 worms were counted for each plate.

### RNA isolation

To isolate RNA, 250 $\mu$l Trizol was added to the worm pellet, vortexed for 30 s and incubated at 4°C with shaking for 1 h. Then 50 $\mu$l chloroform was added, followed by 30 s centrifugation and 3 min at RT. The tubes were then centrifuged for 15 min at 13,000$g$ at 4°C. The top layer (~125 $\mu$l) was transferred to a fresh tube, and the chloroform addition and centrifugation steps were repeated. 125 $\mu$l 2-propanol was added; the tubes were mixed by inverting to avoid shearing the RNA and incubated at RT for 3 min. The tubes were centrifuged at 13,000$g$ for 10 min, and supernatant was carefully decanted without disturbing the RNA pellet. The pellet was washed with 250 $\mu$l 70% RNAse-free ethanol and centrifuged at 18,000$g$ for 5 min at 4°C. Supernatant was removed and the tubes air dried inside the hood. Pellets were dissolved in 10 $\mu$l RNAse-free water and heated at 65°C for 10 min. Concentration was determined using NanoDrop (Thermo Fisher Scientific), and the quality was checked on an agarose gel. Samples were stored and kept at –80°C.

### Quantitative real-time PCR

cDNA was synthesized from 1000 ng of total RNA using iScript cDNA Synthesis Kit (Bio-Rad). The resulting cDNA was diluted 100×, mixed with primer pairs for each gene and SYBRgreen (Bio-Rad). All primer pairs were designed using Primer3 with spliced sequences retrieved from Wormbase (wormbase.org) and were made to span at least two exons. The qRT-PCR reaction was run on a LightCycler 480 (Roche) using 40 cycles, after which the melting curves for each well were determined. Final fold change values were estimated relative to the *act-1* gene in the WT control. List of primers is shown in Table S1.

### Live imaging

Synchronized worms were washed off the plates and washed of bacteria with M9 or sterile water. Slides with 2% agarose in M9 pads were prepared fresh before each imaging. Washed worms were immobilized using imidazole, placed on the agarose pad, and covered with a cover slip secured with nail polish. Imaging was done with temperature-controlled Nikon Ti-E Eclipse inverted/UltraVIEW VoX (Perkin Elmer) spinning disc confocal setup, driven by Volocity software (version 6.3; Perkin Elmer). Unless otherwise specified, images were analysed using ImageJ.

### Mitochondrial morphology

Worms were grown on NGM plates containing 2 $\mu$M Mitotracker Deep Red until L4 stage. After washing, worms were immobilized with imidazole and imaged on a temperature-controlled Nikon Ti-E Eclipse inverted/UltraVIEW VoX (Perkin Elmer) spinning disc confocal setup, driven by Volocity software (version 6.3; Perkin Elmer). Mitochondrial morphology was analysed using ImageJ as described in Valente et al (2017). In short, images were thresholded and skeletonized for the network analysis. Individual mitochondria were analysed manually from raw images with the background removed. Statistical difference between groups was determined using multiple $t$ tests.

### Nile red staining

Nile red stock solution was prepared as a 5 mg/ml solution in acetone. The stock solution was stirred in dark for 2 h before use. Working solution was made by diluting 6 $\mu$l of the stock solution in 1 ml 40% isopropanol. Worms were washed from the plates, and bacteria were removed by washing in PBST. Pellet of washed worms was incubated for 3 min at RT with gentle rocking in 40% isopropanol. After incubation, worms were pelleted and the supernatant removed. Worm pellet was incubated in 600 $\mu$l working solution for 2 h in the dark. After incubation, Nile red solution was removed, and the worm pellet was washed in PBST for 30 min. Worms were imaged on a 2% agarose pad on a Nikon Ti-E Eclipse inverted/UltraVIEW VoX (Perkin Elmer) spinning disc confocal setup, driven by Volocity software (version 6.3; Perkin Elmer). Green fluorescence was examined. Imaging of unstained control under the same conditions gave no signal.

### Oil Red O staining

Oil Red O (ORO) stock solution was prepared as a 5 mg/ml solution in 100% isopropanol. Working solution was made by diluting stock 3:2 in water (60% isopropanol final concentration). The working

solution was filtered through a 0.2-$\mu$m filter and allowed to mix for 2 h before use. Worms were washed from the plates, and bacteria were removed by washing in PBST. Pellet of washed worms was incubated for 3 min at RT with gentle rocking in 40% isopropanol. After incubation, worms are pelleted and the supernatant is removed. The worm pellet is incubated for 2 h in 600 $\mu$l of ORO working solution with gentle mixing. After staining, the pellet is washed for 30 min in PBST. Imaging is done with a color-capable camera.

### ROS measurement

Worms were washed from the bacteria until the supernatant is clear. Washed worms were fixed in 3.7% formaldehyde for 15 min. Fixed worms were pelleted and washed in M9. The worm pellet was stained with 5 $\mu$M Cell ROX Deep Red for 1 h. Stained worms were imaged within 2 h and red fluorescence proportional to the amount of ROS was quantified with ImageJ. Statistical difference was determined using multiple $t$ tests.

### Statistical analysis

Unless otherwise stated, data in graphs are mean ± SD from three biological and three technical replicates. Survival experiments were tested with log-rank (Mantel–Cox) test and distributions with Mann–Whitney test. ANOVA plus post hoc test or unpaired $t$ tests were used for all other measurements, as indicated for individual graphs in figure captions. Graphing and statistical analysis were done using GraphPad Prism and Minitab software. ****$P < 0.0001$; ***$P < 0.001$; **$P < 0.01$; *$P < 0.05$.

## Supplementary Information

## Acknowledgements

A Krisko is supported by the Heisenberg grant from the Deutsche Forschungsgemeinschaft. M Musa was supported by the Mediterranean Institute for Life Sciences and by FEBS and EMBO short-term fellowships. Grant 337327 from the European Research Council to N Raimundo and I Milosevic is supported by the John Black Charitable Foundation, Wellcome Trust Investigator award the John Fell Fund. PA Dionisio, R Casqueiro, I Milosevic, and N Raimundo are supported by H2020 grant 857524 to MIA-Portugal. The authors would like to thank Tea Copić, Dirk Schwitters, and Marina Konta for their excellent technical contribution.

### Author Contributions

M Musa: conceptualization, formal analysis, investigation, visualization, methodology, and writing—original draft.
PA Dionisio: formal analysis, supervision, investigation, methodology, and writing—review and editing.
R Casqueiro: investigation and methodology.

I Milosevic: conceptualization, supervision, funding acquisition, methodology, and writing—original draft.
N Raimundo: conceptualization, formal analysis, supervision, funding acquisition, methodology, and writing—original draft.
A Krisko: conceptualization, formal analysis, supervision, funding acquisition, methodology, and writing—original draft.

### Conflict of Interest Statement

The authors declare that they have no conflicts of interest.

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
