## [Reviewer comments · Life Science Alliance]

Life Science Alliance

Lack of peroxisomal catalase affects heat shock response in *C. elegans*

Marina Musa, Pedro Dionisio, Ricardo Casqueiro, Ira Milosevic, Nuno Raimundo, and Anita Krisko

DOI: <https://doi.org/10.26508/lsa.202201737>

Corresponding author(s): Anita Krisko, Universitätsmedizin Göttingen

Review Timeline:

Submission Date:	2022-09-26
Editorial Decision:	2022-10-18
Revision Received:	2022-10-24
Accepted:	2022-10-24

Transaction Report:

Please note that the manuscript was reviewed at Review Commons and these reports were taken into account in the decision-making process at Life Science Alliance.

Full Revision

Manuscript number: RC-2021-00944

Corresponding author(s): Anita Krisko

[Please use this template only if the submitted manuscript should be considered by the affiliate journal as a full revision in response to the points raised by the reviewers.]

*If you wish to submit a preliminary revision with a revision plan, please use our "Revision Plan" template. **It is important to use the appropriate template to clearly inform the editors of your intentions.**]*

1. General Statements [optional]

This section is optional. Insert here any general statements you wish to make about the goal of the study or about the reviews.

In our study, we aimed to address the role of peroxisomes in the heat shock hormesis and thermotolerance. Peroxisomes are widely studied in the context of oxidative stress and metabolism, however have not received much attention in the context of the stress responses, in particular the heat shock response. We consider our study of potentially wide interest to readers; however, we are aware that this is a mere beginning and that numerous mechanistic steps are yet to be uncovered.

In the name of all authors, I would like to thank to all Reviewers for their valuable feedback, both positive comments, as well as the criticism. We have carefully read and considered all the comments, and have decided to accept all the minor comments and those related to textual changes, which are already included in the revised manuscript. In addition, we carried out all suggested experiments, as detailed below.

This section is mandatory. Please insert a point-by-point reply describing the revisions that were already carried out and included in the transferred manuscript.

Reviewer #1 (Evidence, reproducibility and clarity (Required)):

In the manuscript " Functional peroxisomes are required for heat shock-induced hormesis in *Caenorhabditis elegans*" the authors show that a hormetic heat shock requires the peroxidase catalase *ctl-2* for longevity and thermotolerance. Furthermore the authors characterize the hormetic stress response in *ctl-2* mutants and show that *ctl-2* is required for the proper formation of HSF-1 stress granules, and that *ctl-2* mutants have a changed transcriptional response to heat shock and a changed induction of heat-stress induced oxidative stress. The authors go on to characterize peroxisomes, and peroxidase-regulated fatty acids, as well as mitochondria in both *ctl-2* mutants and upon heat shock. The authors conclude that functional

peroxisomes play an important role in the hormetic heat stress response.

This is an interesting study, that demonstrates that the loss of the peroxidase catalase *ctl-2* plays an important role in the heat stress response. The authors need to provide more details on experimental repeats and experiments strengthening the conclusion that peroxisomes are generally important for the hormetic stress response would also improve this manuscript.

Furthermore, some textual changes would clarify some specifics in this manuscript.

****Major issues:****

1. Throughout the manuscript it is unclear how many times the experiments were conducted (e.g. Lifespan, thermotolerance) or how many biological replicates were used (qRT-PCR, peroxisome characterization etc). This critical information should be included. A table with the individual lifespan experiments and thermotolerance experiments is expected.

We are grateful to the Reviewer for drawing our attention to this and we have now included this information throughout the manuscript.

2. The authors are characterizing the *ctl-2* mutant upon heat stress and find some compelling differences to wild-type animals. The authors state "We measured the expression levels of two peroxisomal transport proteins, PRX-5 and PRX-11, and found no differences in expression between WT and Δ *ctl-2* strains, suggesting that morphogenesis should not be impaired and that proteins needed for their proper function should be present, with the exception of *ctl-2* in the Δ *ctl-2* strain (Figure 3A)." The authors however conclude that functional peroxisomes are required for the benefits of a hormetic HS. What the authors are in fact demonstrating is that *ctl-2* specifically is required for the hormetic HS response. To demonstrate that peroxisomes in general are required they should disrupt peroxisome function by additional means. What the authors also demonstrate is that heat shock has an effect on peroxisomes. These differences need to be clarified. It is also not fully clear whether peroxisomes are functional in *ctl-2* mutants? Further peroxisome-relevant enzymes should be tested for levels and functionality in *ctl-2* mutants and upon HS.

This is correct and we have now addressed the above-mentioned issues throughout the manuscript. As the Reviewer noted, the peroxisomes are functional if monitored as the transcript levels of two peroxisomal transport proteins, implying that they have the potential to import their components. On the other hand, they are lacking the catalase, so they do not have a complete set of their components. Our phrasing in the original manuscript indeed was not optimal and we have now modified the conclusions, stating the peroxisomal catalase is important for the proper heat shock response.

Moreover, it is partially correct to say that a thorough explanation is missing as to what heat shock does to peroxisomes. We would like to emphasize that we measured the transcript levels of three enzymes from the very long chain fatty acid oxidation: straight-chain acyl-CoA oxidase, ACOX-1, MAO-C-like dehydratase domain protein, MAOC-1, and propanoyl-CoA C-acyltransferase, DAF-22. We have now measured the transcript levels of several other peroxisomal enzymes in the WT and the *ctl-2* mutant under optimal conditions and heat shock and reported about it in the manuscript.

3. The TORC1 experiments are too indirect. The authors should perform WesternBlots with a S6Kinase -Phospho antibody to determine whether TORC1 is inhibited or not. Alternatively, the

authors may choose to remove this experiment from the manuscript without disrupting the main message of the manuscript, since no link between TORC1 and peroxisome function nor with HS has been established.

We are aware that the classical experiment to measure the activity of TORC1 is the Western Blot based assay to observe the phosphorylation of S6 kinase. However, we find the available antibodies not good enough to perform this experiment. We, therefore, conducted a different experiment to estimate the TORC1 activity, and it is the one described in our manuscript. We suggest that, as the phosphorylation of S6 is a downstream readout of TORC1 activity, so is the transcript level of the group of genes we measured, as reported previously (Das, Melo et al. 2017; Kenyon 2010; Kenyon 2011; Robida-Stubbs, Glover-Cutter et al. 2012; from the manuscript). While we are aware of the shortcomings of the assay we used, we find it valuable to report that the heat shock has an inhibitory effect on the TORC1 activity in WT *C. elegans* but not in the peroxisomal mutant. We would like to ask the Reviewer to reconsider this comment.

****Textual changes:****

1. The authors should adhere to *C. elegans* nomenclature convention (at least once), by referring to the *ctl-2* deletion with the specific allele name [*ctl-2(xx)*] (unlike in yeast, Δ is not commonly used). Furthermore a more thorough description of the specific allele (complete gene deletion? Point mutation? Truncation?) should be included.

We have now taken care of this issue.

2. Throughout the manuscript the authors should refer back to their specific hormetic heat shock paradigm, since several studies (also cited here) have shown differences in the specific physiological changes (e.g. HS on day 1 of adulthood has profound effects on broodsize, whereas the authors show here that a HS at L4 does not, other differences in terms of activation of other stress reporters has also been reported).

Since our data regarding brood size were not put into context of the present study and were not studied in-depth, we have decided to remove these results from the present manuscript.

3. The abbreviation OGT (original growth temperature??) should be defined

OGT stands for optimal growth temperature and we have now defined it in the manuscript.

4. The authors should change the sentence "The binding of HSF-1 to the HSEs can be visualized by a HSF-1::GFP fusion protein". This is not technically true. While HS has been shown to induce HSF-1 stress granules and these stress granules have been indirectly shown to correlate with transcription, it has however not been demonstrated that HSF-1 is bound to HSEs or whether it is bound to other DNA stretches or what is in fact transcribed by HSF-1 when it localizes to these foci.

We are grateful to the Reviewer for this clarification, which we have included into the manuscript.

5. The authors describe the morphological changes of peroxisomes upon HS, however they

missed the opportunity to fully interpret these results in the discussion. Again, their conclusion that peroxisomes may be impaired in *ctl-2* mutants is very vague.

We agree with this comment and we have now taken care of this issue by introducing additional clarifications.

Reviewer #1 (Significance (Required)):

Hormesis and the role of peroxisomes in stress responses is an important and interesting topic. The role of peroxisomes in stress responses has not been addressed. Researchers with interest in stress responses will be interested in this work.

My expertise lies in *C.elegans* stress and longevity with a specific focus on hormetic mechanisms.

Referee Cross-commenting

While I agree with most comments from the other referees, I don't believe it is feasible to ask the authors to generate *C. Elegans* cell cultures for any follow up experiments. I would be satisfied with a more thorough comparison of the HS response between WT and *Ctl-2* mutants, i.e: compare preconditioning: 1h and 4h HS at L4 and d1 and then do thermotolerance experiments

We thank the Reviewer for this comment. We agree that to generate *C. elegans* cell cultures is not feasible. We have, however, performed the HS treatments in mammalian cells in culture.

HEK cells were transfected with Lipofectamine 3000 for 48h with either negative siRNA control (siCont), siPEX5 or siCAT at 100 nM, according to the manufacturer's protocol. For HSR induction, cells were moved for 1h to a humidified incubator containing 5% CO₂ at 42,5°C, or maintained at 37°C. After, RNA was immediately isolated with NucleoSpin RNA Columns for RNA purification (Macherey-Nagel), and 1000 ng of total RNA was converted to cDNA with iScript™ cDNA Synthesis Kit (Biorad). Relative gene expression of PEX5, catalase (CAT), HSP90AA1, HSP70 (Hspa1b) and small HSPs (Hspb1, Hsph1 and Hspe1) were determined by qPCR, which were performed in a CFX Opus 384 Real-Time PCR System (Biorad) in a final volume of 5 µL using 2× SYBR Green PCR master mix and 0.3 µM of each primer pair. Relative gene expressions were calculated according to the Pfaffl method, with the results being normalized to the three most stable housekeeping genes in this experimental setting (HPRT, B2M and RPL7), as determined with NormFinder. The Ct averages of the four siCont samples at 37°C were used as the calibrators to determine ΔCt for each gene. Data comparisons were conducted with two-way analysis of variance (ANOVA) followed by Bonferroni post hoc tests and data is presented as mean ± SEM (Figures 1 and 2).

Although the siRNA-mediated silencing of CAT and PEX5 was successful (93% and 60% respectively), as determined by qPCR (**Figure 1**), the expression of several HSPs (HSP70, HSP90 and the small HSPs Hspb1, Hsph1 and Hspe1) remained similar following HS between either CAT- or PEX5- silenced cells and siCont-transfected cells (**Figure 2**). Preliminary data from SH-SY5Y cells treated as described above yielded similar results (data not shown). These observations may suggest that catalase and/or functional peroxisomes are not necessary for HSR induction and hormesis, but these experiments were limited to two cell lines and one form of HS (1h at 42,5°C). Therefore, it is possible that different protocols may yet unravel a still undescribed link between peroxisomes, HSR, and life extension, but we to pursue this will take a long time and would be far outside the scope of this manuscript.

Figure 1. siRNA-mediated silencing of CAT and PEX5. Data presented as scatter plot graphs plus mean \pm SEM from five independent experiments. *** $p < 0.001$ vs. respective siCont-treated group at 37°C or 42,5°C. * $p < 0.05$ vs. indicated group.

Figure 2. HSPs expression remained unaffected following HS in CAT- or PEX5-silenced HEK cells. Data presented as scatter plot graphs plus mean \pm SEM from five independent experiments. * $p < 0.05$, ** $p < 0.01$, *** $p < 0.001$ vs. respective siRNA-treated group at 37°C.

Reviewer #2 (Evidence, reproducibility and clarity (Required)):

****Summary:****

In the manuscript entitled "Functional peroxisomes are required for heat shock-induced hormesis in *Caenorhabditis elegans*", Musa et al. explored the role of functional peroxisomes in the heat shock (HS)-induced hormesis in *C. elegans* using peroxisomal catalase *ctl-2* deletion mutants. They showed that *ctl-2* deletion abolished HS-induced longevity and suppressed the HS-induced upregulation of small heat shock proteins and HS-induced HSF-1 nuclear accumulation. Furthermore, the authors analyzed the differences in HS-induced phenotypes between wild type animals and *ctl-2* mutants: the activation of the antioxidant response, the pentose phosphate pathway, and increased triglyceride content, which are the most prominent changes observed during heat shock in wild-type animals. However, there is only weak evidence supporting their claim that functional peroxisomes are required for HS-induced hormesis.

****Major comments:****

1. The authors conclude that the functional peroxisomes are essential for the HS-induced hormesis based only on one observation that the transient heat shock treatment did not increase the lifespan in the short-lived *ctl-2* mutants.

This is only partially correct. In addition to the absence of lifespan extension in the *ctl-2* mutants, our conclusion was based also on the lower extent of the heat shock protein expression during HS in *ctl-2* mutant. In the revision, we have reinforced this conclusion by adding the measurements of the kinetics of the heat shock response activation in the WT and the *ctl-2* mutant, whereby the possible differences in the kinetics between the strains have been excluded as a possible interpretation. The results are now reported in the manuscript.

Because the *ctl-2* mutants are short-lived, the author should have carefully examined the relevant role of functional peroxisomes in hormesis response by conducting lifespan measurements using *ctl-2* RNAi-treated animals or mutants deficient in other genes essential for peroxisome function.

We entirely agree with this comment. Since we are unable to perform such an elaborate study in this moment, and due to other similar comments from other Reviewers, we have decided to modify the conclusion, pointing out that it is the lack of the peroxisomal catalase that is associated to the lower extent of the heat shock response activation.

2. The authors examined the difference between wild-type animals and *ctl-2* mutants in many aspects; however, there is no rational explanation or examination that their observations have a role in HS-induced hormesis.

We acknowledge that we are lacking the mechanistic insight into the reported phenomenon. This will be the focus of future studies, while in our present study, we have modified the conclusions to match the presented results.

3. There is a severe shortage in the explanation of their experiments, especially for the methods of the experiments and statistical procedures. The authors should clarify the details of the experiments with proper statistical analyses. Follows are the points to be clarified for Figure 1 as

insufficiency examples in the explanation in their study.

We have now taken care of this issue throughout the manuscript.

Fig. 1A, B: Is this the representative survival curve of several experiments? The authors should clarify this point. The authors should include a table with the statistical numbers: the number of animals they used, the number of animals they measure in each experiment, and the p-value.

We have now clarified this issue in the manuscript.

Fig. 1C: Is this a representative brood size of one animal, the mean brood size of some animals of one representative experiment, or the mean brood sizes from some experiments? The authors should clarify this point. Also, the authors should perform statistical analysis.

We have removed the results related to the brood size from the present version of the manuscript.

Fig. 1D-F: The authors should state the number of experiments they conducted. The author should perform multiple comparison statistical analyses instead of t-test without any correction.

Done.

Fig. 1H: Is this a representative plot of a representative experiment or the plot of the mean values from some experiments? The authors should clarify this point. The author should also clarify the tissue they analyzed and the exact number of the experiment (the number of nuclei in each animal they analyze, the number of animals they analyze in each experiment). The author should also clarify the detailed statistical procedure: one-way ANOVA or two-way ANOVA; which multiple comparison method they used in their analyses.

We have now clarified all these issues.

****Minor comments:****

The author should explain the details for the experiments in either the Methods section or the figure legends.

The authors should integrate the figures 6B-J (6B-D -> one figure, 6E-G -> one figure, and 6H-J -> one figure) into three figures because there are redundancies in these figures. In addition, the authors should perform the statistical analyses with multiple comparison procedures instead of a simple t-test.

We have now done as suggested by the Reviewer.

The authors should perform statistical analyses on Figs. 6N and 6O.

Done.

The authors would better cite relevant articles when referring to representative target genes of UPRER, UPRmt, and TORC1.

Done.

Reviewer #2 (Significance (Required)):

Understanding the mechanism underlying the HS-induced hormesis in a multicellular organism is essential in the research field. Their finding that functional peroxisomes play a pivotal role in the HS-induced hormesis, if properly demonstrated, would provide us with the significant progress in this field; however, much more proper experiments are required to support their conclusion. Nevertheless, their finding can stimulate the attention of researchers who study the aging process, stress response, and especially peroxisome function.

I am an expertise in the study of aging in *C. elegans*.

Reviewer #3 (Evidence, reproducibility and clarity (Required)):

In this MS, the authors aim at the understanding of the role of peroxisome related-anti-oxidant capacity for the effect of a heat conditioning treatment on the HSR and associated longevity. Despite the finding that *ctl-2* mutants they used show reduced resistance upon heat pre-conditioning, neither the mechanism (*Ctl-2* is required for HSR) nor the claim that functional peroxisomes are required for heat shock-induced hormesis are, in my view, fully proven by the data in this MS. What the data basically show is that, related to a fragile status of the *ctl-2* mutants, pre-conditioning was either to severe (toxic) or/and lead to development defects such that it was no longer effective in priming organismal resistance, likely to HSR-independent features.

****Major comments:****

Figure 1:

The level of pre-conditioning induced resistance (as opposed to intrinsic sensitivity) that can be induced in a given genetic background is dependent on a number of things, one of which is the severity of the priming dose. A more severe heat shock (that initially causes more damage) leads to a slower rate of tolerance development but the level of tolerance (of the surviving cells) is much higher. However, if too toxic, the priming treatment will result in loss of cells, which - at the organismal level- not or less reveal the resistance of the surviving primed and thus tolerant cells.

Agreed.

Furthermore, such intrinsic sensitivity (un-primed) is determined by many more factors than only the capacity to induce the HSR. It is thus important to better evaluate the relevance of differences in intrinsic sensitivity of wildtype and *ctl-2* mutants to the priming heat shock (figure 1F). Albeit interesting that *Ctl-2* strains are hypersensitive to heat, this data also could imply that the real mechanism of being able to build up induced-resistance and longevity is not mechanistically due to an altered regulation of the HSR, but merely a reflection of that intrinsic difference in the sensitivity to the damage inflicted by the priming heat treatment.

Also, as the treatment was given during the L4 stage, many of the effects may be blurred by differences in the sensitivity to heat treatment on developmental processes, conditions under

which also many HSP are differently regulated (also HSF-1 independently). Whilst still interesting, this is even more complex to interpret mechanistically.

We chose the late L4 stage as opposed to adults since worms are mostly developed at L4, and have no embryos, which is a big advantage. The presence of embryos may confound the measurements, especially in the qPCR experiments since they may express HSPs due to their role in developmental processes as opposed to heat stress. It was also reported elsewhere that L4 stage is the most sensitive to HS in terms of HSR, while exhibiting higher survival rate than younger or older worms following HS. We have included this explanation in the manuscript.

In fact, to conclude on the mechanistic involvement on peroxisome redox status for resistance inducing by heat priming, one of would require to e.g. derive cell lines from wildtype and *ctl-2* mutant worms and perform an induced-thermotolerance / survival experiments (a iso-toxic and iso-dose heat priming treatments) to see whether *ctl-2* truly have an impaired HSR due to cell autonomous features.

We are not able to establish cell lines from *C. elegans*, however, we have performed the suggested experiment in the mammalian cells in culture.

HEK cells were transfected with Lipofectamine 3000 for 48h with either negative siRNA control (siCont), siPEX5 or siCAT at 100 nM, according to the manufacturer's protocol. For HSR induction, cells were moved for 1h to a humidified incubator containing 5% CO₂ at 42,5°C, or maintained at 37°C. After, RNA was immediately isolated with NucleoSpin RNA Columns for RNA purification (Macherey-Nagel), and 1000 ng of total RNA was converted to cDNA with iScript™ cDNA Synthesis Kit (Biorad). Relative gene expression of PEX5, catalase (CAT), HSP90AA1, HSP70 (Hspa1b) and small HSPs (Hspb1, Hsph1 and Hspe1) were determined by qPCR, which were performed in a CFX Opus 384 Real-Time PCR System (Biorad) in a final volume of 5 µL using 2× SYBR Green PCR master mix and 0.3 µM of each primer pair. Relative gene expressions were calculated according to the Pfaffl method, with the results being normalized to the three most stable housekeeping genes in this experimental setting (HPRT, B2M and RPL7), as determined with NormFinder. The Ct averages of the four siCont samples at 37°C were used as the calibrators to determine ΔCt for each gene. Data comparisons were conducted with two-way analysis of variance (ANOVA) followed by Bonferroni post hoc tests and data is presented as mean ± SEM (Figures 1 and 2).

Although the siRNA-mediated silencing of CAT and PEX5 was successful (93% and 60% respectively), as determined by qPCR (**Figure 1**), the expression of several HSPs (HSP70, HSP90 and the small HSPs Hspb1, Hsph1 and Hspe1) remained similar following HS between either CAT- or PEX5- silenced cells and siCont-transfected cells (**Figure 2**). Preliminary data from SH-SY5Y cells treated as described above yielded similar results (data not shown). These observations may suggest that catalase and/or functional peroxisomes are not necessary for HSR induction and hormesis, but these experiments were limited to two cell lines and one form of HS (1h at 42,5°C). Therefore, it is possible that different protocols may yet unravel a still undescribed link between peroxisomes, HSR, and life extension, but we to pursue this will take a long time and would be far outside the scope of this manuscript.

Figure 1. siRNA-mediated silencing of CAT and PEX5. Data presented as scatter plot graphs plus mean \pm SEM from five independent experiments. *** $p < 0.001$ vs. respective siCont-treated group at 37°C or 42,5°C. * $p < 0.05$ vs. indicated group.

Figure 2. HSPs expression remained unaffected following HS in CAT- or PEX5-silenced HEK cells. Data presented as scatter plot graphs plus mean \pm SEM from five independent experiments. * $p < 0.05$, ** $p < 0.01$, *** $p < 0.001$ vs. respective siRNA-treated group at 37°C.

Related to the actual data in figure 1A-c, also relative effects need to be taken into account as the Ctl-2 mutants are short lived. E.g., if one looks at maximum life span, the differences

Full Revision

between the strains seem minimal ($20/17 = 1,17$ fold for wildtype and $15/13 = 1,15$ fold for *ctl-2*), so to conclude on no effect of heat pre-conditioning in *ctl-2* strains seems an overstatement.

This is not entirely correct. Usually, in lifespan measurements in *C. elegans*, we compare the median lifespan. For WT worms, we measured increased median and maximum lifespan; median lifespan was increased from 11 to 14 days post HS ($\approx 20\%$ increase), and maximum lifespan from 17 to 20 days (15% increase) (Figure 1A). In contrast, while maximum lifespan of the *ctl-2(ua90)II* strain after HS was increased from 13 to 15 days ($\approx 13\%$ increase), the median lifespan was unchanged and was 10 days for both HS and OGT *ctl-2(ua90)II* worms (Figure 1B). Overall, mild HS did not significantly affect *ctl-2(ua90)II* strain lifespan.

Regarding to the brood size, it is not only true that these are smaller for *ctl-2* worms, but also that there was no effect of the pre-conditioning treatment in wildtype whereas a reduction was caused by the pre-conditioning of the *ctl-2* worms. What does this imply?

Since the results of brood size were not sufficiently understood and out of context of the present study, we have removed them from the current version of the manuscript.

Regarding the HSF/HSP data. First of all, a better visual insight in the quantitative differences in basal transcription levels between the strains should be provided. It looks as if they could be significantly lower for at least HSP16.1 in *ctl-2* strains.

We have now included description of these results in the manuscript.

Next, it would then be essential to evaluate the responses relative to these basal levels in the *ctl-2* lines themselves (and not relative to that in wildtype animals). Second, looking at Hsp70, the HSP being most dependent of HSF1 upon a heat shock, the data imply that there is nothing wrong with the heat shock activated HSF-1 response in *ctl-2* as such. As stated above, magnitude differences might also be a matter of kinetics, so measuring this at a single time point (4h after HS) may e.g. too early for being at its peak in *ctl-2* cells.

We agree with this comment. Therefore, we have performed the experiment where we evaluated the kinetics of the heat shock response in the WT and the *ctl-2* mutant worm. The measurements of the kinetics of the heat shock response activation in the WT and the *ctl-2* mutant reveal that there are no differences in the kinetics between the strains, whereby the possible differences in the kinetics between the strains have been excluded as a possible interpretation. The results are now reported in the manuscript.

Third, it must be emphasized that HSF-1 foci/granules formation is a well-known feature of the response of HSF-1 to heat shock, but these granules are not the site of Hsp transcription, i.e they are not functionally related to HSP expression and not necessarily correlate quantitatively to HSR activation. So, the conclusion that HSF-1 activation/the HSR is truly attenuated in *ctl-2* strains is -in my view- not fully proven. In fact, there is an intricate related between small HSP and oxidative stress and the lower (if correct) Hsp16.1 and Hsp16.2 expression could rather be related to such features.

We thank the Reviewer for drawing our attention to this issue, which we have now addressed in the manuscript by modifying the conclusion of a specific set of results related to the HSF-1 foci

formation during heat shock. We have also discussed more extensively the connection with the oxidative stress.

Figure 2:

As for the HSR, kinetics may be different for wildtype and *ctl-2* strains for all these endpoints, reflecting the higher intrinsic, non-primed heat sensitivity of the *ctl-2* strain. Again, whilst interesting phenotypically and maybe relevant physiologically (i.e. being able to be primed as weak animal to show organismal resistance), this means that the data are elusive in terms of mechanisms.

We acknowledged from the beginning that we are missing mechanistic details of the presented result, and we therefore agree with this comment. As for the possible differences in the kinetics of the heat shock response activation in the WT and the *ctl-2* worms: the measurements of the kinetics of the heat shock response activation in the WT and the *ctl-2* mutant reveal that there are no differences in the kinetics between the strains, whereby the possible differences in the kinetics between the strains have been excluded as a possible interpretation. The results are now reported in the manuscript.

In panel 2E, the lack of elevation in CellROX fluorescence by heat shock in wildtype cells is explained as due to the result in activation of the antioxidant defenses. Whereas this may sound OK, it is contradictory the reasoning given above that heat stress induced oxidative stress and hence cause G6PD upregulation (data panel 2D). In addition, whilst the authors suggest they are the same, to me the CellROX data for *ctl-2* strains appear to be lower (rather than, if anything, higher) for unstressed *ctl-2* strains than wildtype strains. Is this not surprising given they are used as model for an impaired oxidation status? And does this not (also) indicate that the knockout lines have developed compensating strategies? Anyway: I got confused here.

First, there is no significant difference between the CellROX signal of the unstressed WT and the *ctl-2* mutants. Why exactly the WT and *ctl-2* mutant worms have the same CellROX signal is hard to say; the absence of the peroxisomal catalase could have triggered the activation of the cytosolic antioxidant defenses in optimal conditions already, or other compensatory strategies. It is also possible that the absence of the peroxisomal catalase has its consequences only in stressful conditions. In addition, our results during heat shock suggest that the WT strain is able to neutralize the ROS produced during heat shock, unlike the *ctl-2* mutant. The increase in the G6PD expression may have helped with that, since it fed the activation of the pentose phosphate pathway. Our reasoning is that the *ctl-2* mutant was for some reason not able to respond in the same way as the WT, and not that there was no need. The anti-ROS protection seems to have been insufficient during heat shock in the *ctl-2* strain.

Figure 3:

First of all, it seems dangerous to conclude anything on peroxisome NUMBER here as what is measured is the presence of an imported FGP-tagged protein into peroxisomes and hence difference may be (also) due to import related effects. In fact, EM data would be required to make firmer conclusions on peroxisomal morphology, size and numbers.

In principle we agree with this comment, however, we demonstrate in Figure 3 that Prx-5 and Prx-11 expression levels do not display any differences in the WT and *ctl-2* worms. Therefore, we remain confident about the analysis of the peroxisome number. We hesitate to perform an EM analysis due to the lack of specific markers; anything even slightly resembling a peroxisome

could falsely be counted as one. In other words, every methodological approach comes with its own imperfections.

In panel C, it is unclear what is significantly different from what; is the signal in *ctl-2* strains truly lower in *ctl-2* strains than in wildtype strains?

We thank the Reviewer for drawing our attention to this. In Figure 3C, for the significance analysis, everything was compared to the WT in optimal growth conditions. We have now clarified this in the text. The difference between the *ctl-2* mutant and the WT is debatable: while the medians of the data sets are significantly different, this difference may not have any biological significance. Still, we reported the result of the statistical analysis.

Figure 4 - 6: I sympathize with the comprehensive analysis presented in these figures. It is clear to me that different things are either (not) up or down in unprimed *ctl-2* strains and that heat shock does or does not cause similar effects on these endpoints in wildtype and *ctl-2* strains. Whilst this indeed shows that they do respond differently, I do not understand from these that what it all means and, in particular if and how it related causally or consequentially to the impaired priming effects (if true) of the heat shock in *ctl-2* strains.

The results presented in Figure 4-6 have not been put into a mechanistic context in the present study. In the mentioned figures, we do not report any causal or consequential relationships, but we do wish to report on the differential phenotypes of lipid metabolism and storage, as well as mitochondrial morphology in the WT and *ctl-2* mutant worms. However, the presented analyses were performed well and we believe that these results may benefit to other researchers working in related topics.

****Minor additional comments (textual only)****

Whereas this paper discusses the possibility of pre-heat conditioning to induce (long term) resistance, the generality of this as being a hormesis response (or general stress responses) related to other challenges is not warranted and all text concerning that should be deleted. Stresses damaging primarily DNA (ionizing radiation) or proteins & lipid (heat shock) are fundamentally different and each of them requires entirely different and largely independent systems to respond to.

Done.

Moreover, whilst the induced HSR is clearly an established hermetic response, this is still far less clear for e.g. DNA damage responses. Therefore, it is also relevant to define the types of stress to which is referred to e.g. when mentioning the environmental stimuli to which *ctl-2* mutants are apparently hypersensitive.

Done.

Also, the text related to cell non-autonomous response is irrelevant to this study and should be deleted.

Done.

Page 3 lines 1-4: It is incorrect to write that the role HSP has been put forward as most relevant to heat-induced hormesis. This is how they were discovered, but it is now clear that they play a

role in many other pre-conditioning induced resistant phenotypes as well as in the cell-intrinsic sensitivity to proteotoxic stresses in general. Please also be aware that for unprimed, intrinsic resistance to e.g. heat shock, pre-existing levels of HSP are more relevant than the ability to activate the HSR. Activating the HSR is more relevant to resistance to more chronic temperature elevations and acquired resistance via the priming (hormesis).

We thank the Reviewer for drawing our attention to this. We agree about the relevance of these points and have modified the manuscript accordingly.

Reviewer #3 (Significance (Required)):

Regulation of the cell intrinsic heat shock response is an important item to understand how cells may be primed to become resilient to (certain) other stresses.

Besides the main studied regulator (HSF-1), many other levels of regulation likely exist and intra-organellar communication and proteostasis might be an important aspect for controlling such regulation.

As such, peroxisome proteostasis (that, unlike other organelles, are not (also) controlling a organellar unfolded protein response) is an interesting organelle that could co-control the cytosolic heat shock response. So, the aim of this study per se is quite interesting, However, although for evaluation of physiological relevance *C. elegans* is a good model of choice, for the mechanistic studies, cellular experiments would have been better suited.

October 18, 2022

RE: Life Science Alliance Manuscript #LSA-2022-01737

Anita Krisko
Department of Experimental Neurodegeneration, University Medical Center Goettingen
Unknown
Goettingen 37073
Germany

Dear Dr. Krisko,

Thank you for submitting your revised manuscript entitled "Lack of peroxisomal catalase affects heat shock response in *C. elegans*". We would be happy to publish your paper in Life Science Alliance pending final revisions necessary to meet our formatting guidelines.

- please address Reviewer 2's remaining points
- please upload your main and supplementary figures as single files and make sure that table files are uploaded as editable doc or excel files, or they are included in the doc file of your main manuscript text
- please add a Running Title, Alternate Abstract/Summary blurb, and a category for your manuscript to our system
- please add the Twitter handle of your host institute/organization as well as your own or/and one of the authors in our system
- please use the [10 author names, et al.] format in your references (i.e. limit the author names to the first 10)
- please add the author contributions and a conflict of interest statement to the main manuscript text
- please double-check your figure legend for Figure 1; it seems as if the panels might be incorrectly labeled

A. FINAL FILES:

B. MANUSCRIPT ORGANIZATION AND FORMATTING:

Sincerely,

Reviewer #1 (Comments to the Authors (Required)):

The authors have toned down their conclusions were necessary.

Reviewer #2 (Comments to the Authors (Required)):

In the manuscript "Lack of peroxisomal catalase affects heat shock response in *C. elegans*," Musa et al. explored the role of peroxisomal catalase CTL-2 in the heat shock process using *ctl-2* mutant in *C. elegans*. They first showed that *ctl-2* deletion abrogated the longevity, upregulation of heat shock proteins, and HSF-1 nuclear accumulation induced by heat shock treatment. They also found a series of cellular responses caused by heat shock treatment in which CTL-2 might be involved. The authors have satisfactorily addressed most of the concerns raised by reviewers. However, their statistical statement is not detailed enough for a clear understanding. The authors should state these points accordingly before published in Life Science Alliance.

Specific comment:

They should state the following in the manuscript.

Do they use one-way ANOVA or two-way ANOVA in each of the experiments?

What method did they use as ANOVA post hoc analyses?

Response to Reviewers**Reviewer #1 (Comments to the Authors (Required)):**

The authors have toned down their conclusions were necessary.

We are grateful to the Reviewer for recognizing the improvements.

Reviewer #2 (Comments to the Authors (Required)):

In the manuscript "Lack of peroxisomal catalase affects heat shock response in *C. elegans*," Musa et al. explored the role of peroxisomal catalase CTL-2 in the heat shock process using *ctl-2* mutant in *C. elegans*. They first showed that *ctl-2* deletion abrogated the longevity, upregulation of heat shock proteins, and HSF-1 nuclear accumulation induced by heat shock treatment. They also found a series of cellular responses caused by heat shock treatment in which CTL-2 might be involved. The authors have satisfactorily addressed most of the concerns raised by reviewers. However, their statistical statement is not detailed enough for a clear understanding. The authors should state these points accordingly before published in Life Science Alliance.

Specific comment:

They should state the following in the manuscript.

Do they use one-way ANOVA or two-way ANOVA in each of the experiments?

What method did they use as ANOVA post hoc analyses?

We have taken care of this issue by indicating in the Methods and in Figure captions whether we used one-way or two-way ANOVA as well as the post hoc test.

October 24, 2022

RE: Life Science Alliance Manuscript #LSA-2022-01737R

Dr. Anita Krisko
Universitätsmedizin Göttingen
Department of Experimental Neurodegeneration
Waldweg 33
Goettingen 37073
Germany

Dear Dr. Krisko,

Thank you for submitting your Research Article entitled "Lack of peroxisomal catalase affects heat shock response in *C. elegans*". It is a pleasure to let you know that your manuscript is now accepted for publication in Life Science Alliance. Congratulations on this interesting work.

DISTRIBUTION OF MATERIALS:

Again, congratulations on a very nice paper. I hope you found the review process to be constructive and are pleased with how the manuscript was handled editorially. We look forward to future exciting submissions from your lab.

Sincerely,
